# A histone demethylase links the loss of plasticity to nongenetic inheritance and morphological change

Nicholas A. Levis [1] & Erik J. Ragsdale [1]

Plasticity is a widespread feature of development, enabling phenotypic change based on the environment. Although the evolutionary loss of plasticity has been linked both theoretically and empirically to increased rates of phenotypic diversification, molecular insights into how this process might unfold are generally lacking. Here, we show that a regulator of nongenetic inheritance links evolutionary loss of plasticity in nature to changes in plasticity and morphology as selected in the laboratory. Across nematodes of Diplogastridae, which ancestrally had a polyphenism, or discrete plasticity, in their feeding morphology, we use molecular evolutionary analyses to screen for change associated with independent losses of plasticity. Having inferred a set of ancestrally polyphenism-biased genes from phylogenetically informed gene-knockouts and gene-expression comparisons, selection signatures associated with plasticity's loss identify the histone H3K4 di/mono-demethylase gene *spr-5/LSD1/KDM1A*. Manipulations of this gene affect both sensitivity and variation in plastic morphologies, and artificial selection of manipulated lines drive multigenerational shifts in these phenotypes. Our findings thus give mechanistic insight into how traits are modified as they traverse the continuum of greater to lesser environmental sensitivity.

A trait's development can shift from being environmentally sensitive (plastic) to insensitive (constitutive)—a process termed genetic assimilation—thereby fixing what was at first a conditionally expressed trait[1–3]. Because both theory and empirical studies have linked assimilation to phenotypic diversification[4–8], the molecular details of plasticity's loss should inform its potential evolutionary impacts. In principle, inference from cases of assimilation in nature could reveal the molecular mechanisms that mediate shifts in developmental plasticity and associated changes in morphology. The identification of such a mechanism would then allow functional tests of whether and how it drives change in a trait's plasticity. Here, we infer and test such a mechanism in a family of nematodes (Diplogastridae) whose common ancestor had resource polyphenism, or discrete plasticity in its feeding morphology and niche use, with several lineages having lost the polyphenism independently[6] (Fig. 1A). As a result, we (i) identify genes associated with loss of polyphenism using a combination of transcriptomic and comparative genomic analyses, (ii) functionally validate the role of a candidate gene in plastic development, and, motivated by the gene's identity, (iii) test the possibility that the loss of its function allows a rapid response to selection.

## Results

### A shared set of genes is polyphenism-biased across *Pristionchus*

In Diplogastridae, the origin of the polyphenism correlated with increased evolutionary rates for mouth morphology, with even higher rates following in lineages where plasticity was lost and a single morph assimilated[6]. We hypothesized that, in this nematode family, genes whose expression was once polyphenism-biased should bear a signature of evolution (*e.g.*, relaxed or positive selection) associated with plasticity's loss[4,9,10]. To test this idea, we first inferred a set of polyphenism-biased genes ancestral to a clade of >50 known species (the genus *Pristionchus*). For this inference, we identified genes whose

[1]Department of Biology, Indiana Universitys, Bloomington, IN 47405, USA. ✉e-mail: nicholasalevis@gmail.com; ragsdale@indiana.edu

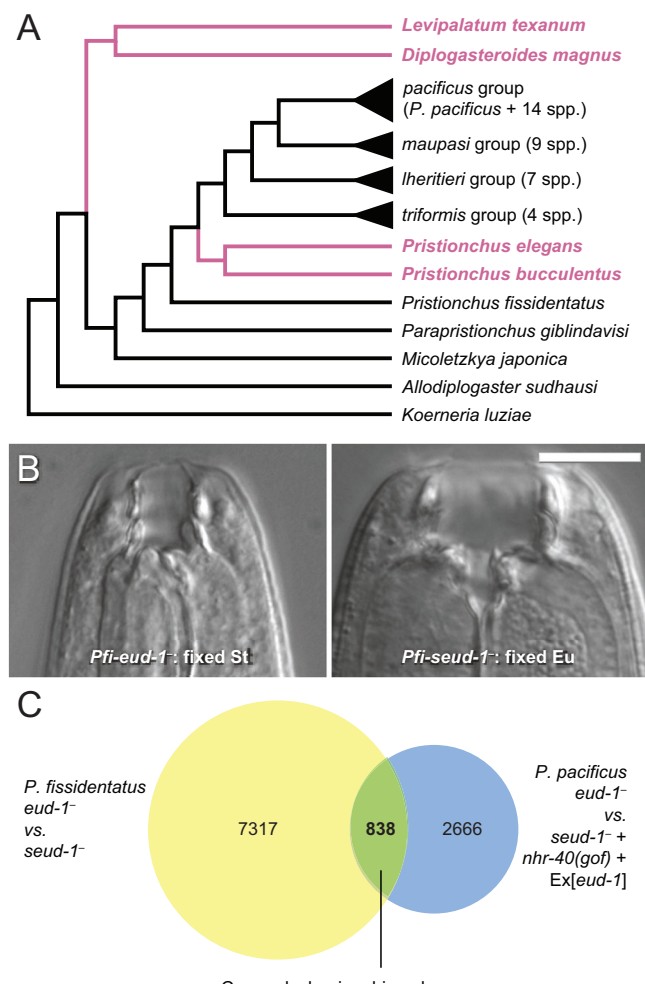

**Fig. 1 | Diplogastrid nematodes for comparing mechanisms of plasticity evolution. A** In a nematode family in which polyphenism in feeding morphology evolved once and persisted across several lineages (black), some lineages (pink) independently assimilated a single morph. Phylogeny is summarized from those previously inferred[6,21]. **B** In *P. fissidentatus* (*Pfi*), ablation of putative switch genes fix alternative forms of the polyphenism. 300 individuals were observed per mutant with similar binary phenotypic output. Scale bar, 10 µm. **C** RNA-seq on constitutively Eu and St lines in *P. fissidentatus* (yellow) and *P. pacificus* (blue) revealed a core set of genes exhibiting the same morph bias across a deep divergence in *Pristionchus* (green). Morph-biased genes for *P. pacificus* were inferred previously[62,63].

expression is changed by the ablation of the polyphenism switch in two species representing an early divergence in that group. Our approach used genetic perturbations in common genetic backgrounds and environmental conditions, which allowed us to distinguish genes associated with the polyphenism per se from others whose expression might also be variously influenced by environmental cues[11]. This approach also excluded effects that might be idiosyncratic to individual cues, which in *Pristionchus* include crowding, starvation, and other metabolic influences[12–14]. Further, these genes were identified from pooled, whole-organism contrasts that, although they obscure responses specific to tissue or developmental stage, we predicted to capture effects across the body, which are known for this case of resource polyphenism[15]. Starting from a known network of switch genes in the mouth-polyphenic model *P. pacificus*, we established a proof of principle that a similar switch mechanism could be found and manipulated in other diplogastrid nematodes. We adapted CRISPR/Cas9-mediated knockout of two polyphenism switch-genes, *eud-1* and *seud-1/sult-1*, in the dimorphic species *P. fissidentatus* (Supplementary

Fig. 1). These knockouts completely fixed the nematodes' feeding phenotypes. Just as in *P. pacificus*[16,17], mutant lines were constitutive for the stenostomatous (St, microbivorous) morph and the eurystomatous (Eu, facultatively predatory) morph, respectively (Fig. 1B; Supplementary Fig. 2). Having confirmed these genes as regulators of the polyphenism across *Pristionchus*, we sequenced the whole transcriptomes of the mutant lines to identify genes with concordant, morph-biased expression in both *P. pacificus* and *P. fissidentatus*. Overlap in morph-biased gene expression in the two species revealed 838 genes (488 St-biased; 350 Eu-biased), defining a core set of polyphenism-biased genes inferred to be ancestral to *Pristionchus* (Fig. 1C).

## Independent bouts of genetic assimilation share common evolutionary targets

We next determined if the evolution of any of these morph-biased genes correlated with independent losses of plasticity. To do so, we first used 41 *Pristionchus* and outgroup species to identify genes uniquely evolving in two species, *P. bucculentus* and *P. elegans*, which have assimilated a single morph[18,19] (Fig. 1A). Although the sister-group relationship between these two species implies a single assimilation event, *P. bucculentus* appears to have fixed the Eu form, whereas the divergence of form in *P. elegans* is so extreme as to confound homology with either the Eu or St morph[20], consistent with high post-assimilation evolutionary rates in Diplogastridae[6]. Therefore, we analysed both Eu- and St-biased genes without assuming which set was constitutively expressed during assimilation. Comparing previously published, whole transcriptomes from across *Pristionchus* and outgroups[21], we identified those genes showing episodic, or branch-specific, diversifying selection in these two species, thereby defining evolutionary targets associated with the loss of plasticity (Supplementary Table 1). We then tested whether any of these genes likewise reported selection associated with an independent assimilation event (Fig. 1A), which would point to a potentially generalizable feature of plasticity's loss in Diplogastridae. Specifically, we intersected our results with patterns of episodic, diversifying selection in the species *Diplogasteroides magnus* and *Levipalatum texanum*, whose mouth morphologies were, like the two *Pristionchus* species above, the product of increased evolutionary rates following assimilation. As in *P. elegans*, the assimilated forms of *D. magnus* and *L. texanum* have diverged beyond identification as either the Eu or St morph, although an outgroup to these species (*Oigolaimella* spp.)[6] suggests the Eu morph may have been fixed. For these analyses, we used these two monomorphic species and polyphenic representatives from other genera with sequenced genomes (*Pristionchus*, *Parapristionchus*, *Micoletzkya*, *Allodiplogaster*, and *Koerneria*) across the phylogeny of Diplogastridae. Whereas our former analysis focused on closely related species within the genus for which the polyphenism-associated genes were inferred (*Pristionchus*), this analysis used distant species from across the family to explore the generalizability of the former's results. Although polyphenism-associated genes were not over-represented among genes showing evidence of selection (Supplementary Table 1), 20 genes reported diversifying selection associated with both assimilation events (Table 1). Therefore, this analysis identified a set of candidates whose function may contribute to plasticity regulation, morphological variation, or both.

To further filter this set of candidates, we tested if the strength of selection acting on genes in each of our assimilated species has been relaxed relative to that on polyphenic species. Our rationale was threefold: first, plasticity can be lost due to selection or drift[3,22–24]; second, genes not associated with the phenotype that was canalized during genetic assimilation might have experienced relaxed selection following plasticity's loss[4,10]; third, this analysis would identify additional targets whose evolutionary history is similar during independent cases of assimilation. Further, because this analysis is not mutually

**Table 1 | Shared polyphenism targets of episodic, diversifying selection during independent bouts of genetic assimilation**

| Morph bias | Species | *P. pacificus* orthologue | Best *C. elegans* match |
|---|---|---|---|
| Eu | *Pbu + Dmag* | ppa_stranded_DN4514_c0_g1_i1 | *C53B4.4b* |
| Eu | *Pbu + Dmag* | PPA00535 | *H11E01.3* |
| Eu | *Pel + Dmag* | Contig11-snapTAU.554 | *F23H11.4a* |
| Eu | *Pel + Ltex* | ppa_stranded_DN35860_c0_g1_i1 | *cope-1* |
| Eu | *Pbu + Pel + Dmag* | ppa_stranded_DN23715_c0_g1_i6 | *Y69A2AR.16a* |
| Eu | *Pbu + Pel + Dmag* | ppa_stranded_DN29329_c0_g1_i3 | *F22G12.5c* |
| Eu | *Pbu + Pel + Dmag* | ppa_stranded_DN8012_c0_g1_i1 | *klp-15* |
| Eu | *Pbu + Pel + Ltex* | ppa_stranded_DN16305_c0_g1_i1 | *spr-5* |
| Eu | *Pbu + Pel + Ltex* | PPA17337 | *unc-116* |
| Eu | *Pbu + Pel + Ltex* | PPA25430 | *pat-3* |
| Eu | *Pel + Dmag + Ltex* | PPA12726 | *T23E7.2a* |
| Eu | *Pel + Dmag + Ltex* | PPA36936 | *fbp-1* |
| Eu | *Pbu + Pel + Dmag + Ltex* | Iso_D.13320.1 | *bli-4* |
| St | *Pbu + Dmag* | ppa_stranded_DN27733_c2_g1_i1 | *nprl-3* |
| St | *Pbu + Dmag* | ppa_stranded_DN28307_c0_g3_i1 | *aakg-4* |
| St | *Pel + Dmag* | ppa_stranded_DN29646_c0_g1_i2 | *C05D11.8* |
| St | *Pbu + Pel + Dmag* | Iso_D.14127.1 | *let-526* |
| St | *Pbu + Pel + Ltex* | ppa_stranded_DN479_c0_g2_i1 | *n/a* |
| St | *Pbu + Dmag + Ltex* | PPA17454 | *ncx-1* |
| St | *Pel + Dmag + Ltex* | ppa_stranded_DN31001_c0_g2_i1 | *smg-6* |

Morph associations are based on the designations of *Pristionchus pacificus* orthologues, and *Caenorhabditis elegans* gene names represent best BLAST hits in that species. *Dmag, Diplogasteroides magnus; Ltex, Levipalatum texanum; Pbu, P. bucculentus; Pel, P. elegans.*

**Table 2 | Shared polyphenism targets of weakened selection during independent bouts of genetic assimilation**

| Morph bias | Species | *P. pacificus* orthologue(s) | Best *C. elegans* match |
|---|---|---|---|
| Eu | *Pbu + Dmag + Ltex* | PPA15798 | *C05G5.1* |
| Eu | *Pbu + Ltex* | PPA00535 | *H11E01.3* |
| Eu | *Pbu + Ltex* | PPA19644 | *obr-3* |
| St | *Pbu + Ltex* | ppa_stranded_DN29992_c1_g3_i2 | *wht-2* |
| St | *Pbu + Ltex* | Iso_D.6849.1 | *cra-1* |
| Eu | *Pbu + Pel + Dmag + Ltex* | ppa_stranded_DN30992_c1_g8_i5 | *wdr-20* |
| Eu | *Pbu + Pel + Dmag + Ltex* | isototal.2131.1 | *hrde-1* |
| St | *Pbu + Pel + Dmag + Ltex* | PPA33928, PPA02058 | *C05E7.1 (tag-196)* |
| Eu | *Pbu + Pel + Ltex* | PPA24448 | *F14H12.3* |
| Eu | *Pbu + Pel + Ltex* | ppa_stranded_DN16305_c0_g1_i1 | *spr-5* |
| Eu | *Pbu + Pel + Ltex* | PPA25430 | *pat-3* |
| Eu | *Pbu + Pel + Ltex* | ppa_stranded_DN28307_c0_g3_i1 | *aakg-4* |
| St | *Pbu + Pel + Ltex* | PPA08312 | *B0563.6a* |
| Eu | *Pel + Dmag* | ppa_stranded_DN29153_c1_g1_i5 | *aagr-4* |
| Eu | *Pel + Dmag + Ltex* | ppa_stranded_DN29329_c0_g1_i3 | *F22G12.5c* |
| St | *Pel + Dmag + Ltex* | ppa_stranded_DN8904_c0_g1_i1 | *tbcd-1* |
| St | *Pel + Dmag + Ltex* | ppa_stranded_DN25492_c0_g3_i2 | *ZC376.3* |
| Eu | *Pel + Ltex* | ppa_stranded_DN22007_c0_g1_i1 | *n/a* |

Morph associations are based on the designations of *Pristionchus pacificus* orthologues, and *Caenorhabditis elegans* gene names represent best BLAST hits in that species. *Dmag, Diplogasteroides magnus; Ltex, Levipalatum texanum; Pbu, P. bucculentus; Pel, P. elegans.* In one case, the best match was ambiguous, so both candidate matches are listed.

exclusive with that for episodic diversifying selection above, given their different methods of detecting increased ω ratios, it is also possible that the two analyses would point to a set of overlapping genes. Therefore, we predicted our combined approach to reveal genes accumulating variation under both kinds of selection. As with our inference of episodic diversifying selection, we observed a limited set of once morph-biased genes (18 total) with signatures of relaxed selection across both assimilation events (Table 2). Moreover, five of these genes were common to both selection analyses. Because the latter genes showed a signature of multiple evolutionary forces, we highlighted these as candidates to inform the process of plasticity's loss and associated morphological change.

## Multiple signatures of selection distinguish the histone demethylase SPR-5

Among the five targets from both selection analyses at both phylogenetic levels was the histone H3-di/monomethyl-lysine-4 (H3K4me1/2) demethylase gene *spr-5/LSD1/KDM1A*. In the model nematode *Caenorhabditis elegans*, *spr-5* is well-established as a mediator of transgenerational, epigenetic inheritance[25–28]. Because (i) *spr-5* has Eu-biased expression, (ii) H3K4 dimethylation is associated with mouth polyphenism in *P. pacificus*[29], (iii) chromatin regulators may act as evolutionary capacitors[30], (iv) transgenerational, epigenetic inheritance has been theoretically implicated in genetic assimilation more generally[22,31–33], and (v) adaptive peak shifts may occur through genetic drift of genes controlling epigenetic variability[22], we investigated *spr-5* as a potential modifier of plasticity. From the sequences reporting selection, we determined whether lineage-specific variants might reveal obvious changes in SPR-5 function. Specifically, we compared the predicted functional consequences between candidate amino-acid sites experiencing diversifying selection and other sites diverging from a sequence reconstructed for the most recent common ancestor of *P. bucculentus* and *P. elegans*. We found no bias between variants reporting diversifying selection and other variants across functional domains of the protein (Table 3). Additionally, there were no major differences in mutational sensitivity or the presence of inferred catalytic or binding motifs between the two site categories, nor was there strong evidence for any single aspect of protein function to be targeted or modified. Together with our selection tests above, these results suggest that relaxed selective constraint potentially shaped much of this gene's divergence or concomitant shift in function and that such relaxation could promote substantial changes in plasticity and morphology.

## SPR-5 regulates polyphenism switching and plastic morphology

We then directly tested the function of *spr-5* in the mouth polyphenism using the polyphenic model nematode *P. pacificus*. When we genetically ablated *spr-5* in *P. pacificus*, we observed two effects on the polyphenism. First, *spr-5* mutants showed decreased plasticity under multiple induction cues. Amidst conditions that alternatively promote the St and Eu morph, specifically liquid and solid rearing media, respectively[14], *spr-5* mutants showed smaller differences in morph bias (estimate = 2.13, $Z = 4.55$, $P = 1.06 \times 10^{-5}$; Fig. 2A) than did wild-type individuals (estimate = 7.37, $Z = 6.65$, $P = 5.92 \times 10^{-11}$). We also tested plasticity in response to an ecologically more relevant cue, starvation. As with culture conditions above, there was a significant genotype-by-environment interaction: when we starved a normally St-biased strain (RS5200B)[16] with and

without a *spr-5* loss-of-function allele, induction of the Eu morph, which was observed in *spr-5* wild-type individuals (estimate = 2.78, $Z = 7.19$, $P = 1.34 \times 10^{-12}$), was abolished in mutants (estimate = −0.48, $Z = 1.91$, $P = 0.11$; Fig. 2B). Together, these results indicate a role for *spr-5* in polyphenism regulation, specifically in environmental responsiveness and induction of Eu morphology. Second, some mutant individuals showed new variants of form, as quantified by geometric morphometrics, suggesting the potential of *spr-5* to mediate change in the plastic morphology itself (Fig. 2C–E). Although the observed morphs of *spr-5* mutants partially overlapped with both wild-type morphs, morphospace was narrower between alternative mutant morphs, as assigned by model-based clustering, indicating an increase of non-stereotypical, intermediate forms. Specifically, morphospace distance between the Eu-like and St-like groups of *spr-5(iub33)* (0.156 units) was significantly shorter (i.e., containing more intermediates) than that between wild-type morphs, including when quantified from representative isolates across *P. pacificus* (0.255 units; $Z = 4.23$, $P = 1.00 \times 10^{-4}$). In short, knocking out *spr-5* affected both the incidence of discrete morphs and their morphology. More generally, mutations in this gene affected the regulation and production of a morphology that has rapidly diversified in correlation with its genetic assimilation.

## *spr-5* influences mouth morphology in the absence of plasticity

Because *spr-5* influences plasticity, we examined whether observed changes to typical mouth morphology were due only to modifications of the polyphenism switch or also to effects downstream of the switch. We reasoned that if *spr-5* could affect once-plastic morphology even after genetic assimilation, *spr-5* mutants should have an effect in *P. pacificus* individuals with morph-constitutive development. To test this idea, we forced fixation of the Eu morph, using a transgenic insertion line that overexpresses *eud-1* (allele *iubIs16*), in a *spr-5* mutant background. We found that despite the absence of plasticity in the transgenic line, *spr-5* mutants showed a different range of morphologies of the Eu morph ($Z = 2.47$, $P = 8.00 \times 10^{-3}$, Procrustes ANOVA; Fig. 3). Thus, in addition to the decreased plasticity in *spr-5* mutants, *spr-5* influences the trait either downstream of or in parallel to the polyphenism network. This finding supports the possibility that altered H3K4 di/monomethylation could result in new variants in mouth morphology both during and after the loss of plasticity.

## Effects of *spr-5* on polyphenism are intergenerational

Given the role of *spr-5* in nongenetic inheritance, we next determined whether mutations in this gene influence polyphenism induction intergenerationally. If so, we would expect the offspring of *spr-5⁻* parents to exhibit the mutant phenotype even after *spr-5* function is restored. To test this, we outcrossed *spr-5⁻* individuals to wild-type (PS312) individuals, and then compared the bias in morph induction among homozygous mutants, heterozygotes, and the wild-type (Fig. 4A). We found that (i) heterozygous $F_1$ showed a significant reduction in the Eu morph compared to a wild-type self-cross, (ii) this intergenerational effect was erased in the $F_2$, which returned to the wild-type phenotype, and (iii) homozygous *spr-5⁻* $F_3$ clones were significantly less Eu-biased than homozygous wild-type $F_3$, showing conversion to the mutant phenotype in exactly one generation. Together, these findings show that the wild-type *spr-5* allele is completely dominant for polyphenism function and that, consistent with the gene's expression in the germline[25,34], *spr-5* regulates plasticity intergenerationally. Further, these results show that, in contrast with phenotypes studied in *C. elegans*[25–27,35], the influence of *spr-5* on the mouth plasticity is immediate, not requiring the accumulation of histone modifications over several generations. Next, to test our presumption that *spr-5* mutant phenotypes are the result of inherited methyl marks, we pharmacologically inhibited SPR-5 activity in wild-type individuals, using a specific inhibitor (Bizine) of human LSD1. Treatment with Bizine phenocopied *spr-5* mutants (Supplementary Fig. 3), both in

**Table 3 | Distribution of amino acid changes across *spr-5* domains for sites with and without evidence of diversifying selection**

| Domain | *P. bucculentus* diversifying variants | *P. bucculentus* other variants | $\chi^2$ | $P$ |
|---|---|---|---|---|
| SWIRM | 0.06 | 0.08 | 2.61 | 0.46 |
| AOL | 0.44 | 0.41 | | |
| SB | 0.44 | 0.30 | | |
| Tower | 0.06 | 0.21 | | |
| Domain | *P. elegans* diversifying variants | *P. elegans* other variants | $\chi^2$ | $P$ |
| SWIRM | 0.67 | 0.00 | 2.71 | 0.44 |
| AOL | 0.00 | 0.36 | | |
| SB | 0.33 | 0.45 | | |
| Tower | 0.00 | 0.18 | | |

Values represent the proportion of each site category that falls within a given domain of the *spr-5* protein and statistics correspond to two-sided test results. Domain identities are based on the crystal structure of the SPR-5 ortholog LSD1 as solved[87].

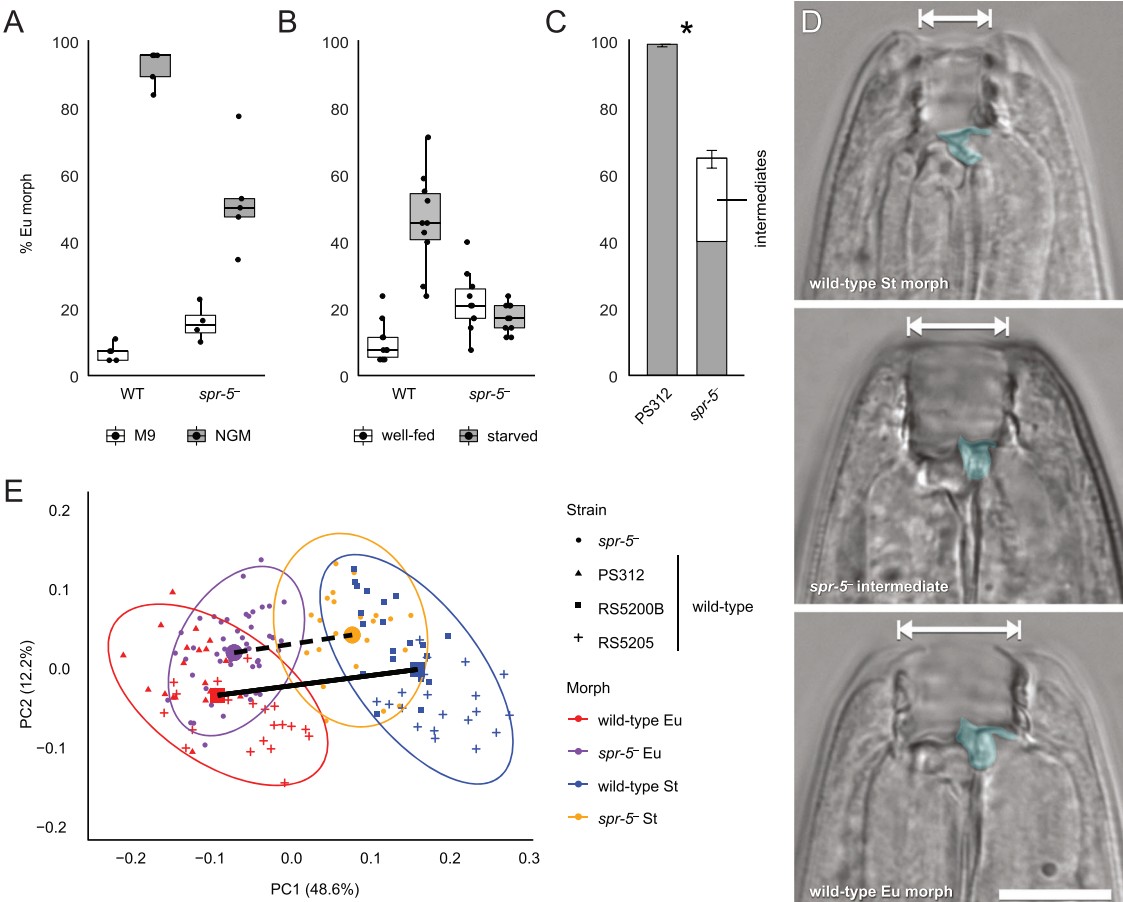

**Fig. 2 | Loss of *spr-5* function affects developmental plasticity and plastic morphology.** Decreased plasticity of *spr-5* mutants in response to alternative morph-induction cues: culture media, in the wild-type (WT) strain PS312 (**A**) and starvation, in the wild-type strain RS5200B (**B**). Putative intermediates, found in *spr-5* mutants in the solid media treatment, were conservatively grouped with St for statistical tests. In (**A**) and (**B**), five biological replicates (solid points) of 30 individuals were phenotyped per treatment group, except for *spr-5* mutants in M9 where *n* = 4. The lower and upper hinges of the box plots correspond to the first and third quartiles, the whiskers extend to the largest value no further than 1.5x the inter-quartile range from the hinges, and the centre corresponds to the median. **C** Proportion of individuals showing the Eu morph, when raised in solid media, after ~10 generations of *spr-5* loss-of-function. This proportion is significantly lower in mutants whether putative intermediates in form are grouped with Eu (estimate = 3.44, $Z$ = 7.33, $P$ = 2.23 × 10$^{-13}$) or St (estimate = 4.51, $Z$ = 9.68, $P$ = 2.00 × 10$^{-16}$)

according to two-sided logistic regression. Phenotypes of the wild-type were scored using five biological replicates of 60 individuals each (*n* = 300 individuals total) and a comparable sample size (*n* = 302 individuals total) for *spr-5* mutants required 13 biological replicates where the number of samples varied per replicate (*n* = 12, 13, 15, 15, 16, 19, 19, 20, 25, 30, 31, 40, 47). Samples were grouped in this panel to highlight the overall proportion of intermediate morphologies observed. Bar plots represent the mean values ± the standard error of the mean. **D** Images of *P. pacificus* natural isolates (top, RS5200B; bottom, PS312) and a *spr-5(iub33)* mutant (centre). Width of mouth (arrow) and shape of dorsal tooth (false-coloured), hallmarks of the *P. pacificus* mouth polyphenism, show intermediate forms in mutants. Scale bar, 10 μm. **E** PCA of geometric morphometrics of *spr-5(iub33)* and wild-type *P. pacificus* morphs. Dashed and solid lines visualize this difference by connecting centroids of mutant and wild-type morphs, respectively. Source data for (**A**), (**B**), (**C**), and (**E**) are provided as a Source Data file.

shifting the morph bias (estimate = 1.32, $Z$ = 3.68 $P$ = 2.33 × 10$^{-4}$) and in the production of otherwise absent, intermediate forms (96 in Bizine treatment, zero in control; $P$ = 2.20 × 10$^{-16}$, Fisher's exact), consistent with the activity of *P. pacificus* SPR-5 in H3K4 di/monodemethylation. Together, these results show that SPR-5 influences, likely through the demethylation of H3K4 marks, intergenerational transmission of polyphenism induction.

## SPR-5-mediated variation fuels multigenerational shifts in plasticity and morphology

Finally, we tested whether loss of *spr-5* function, and presumably the production of epimutations[25–27], would allow directional selection on plasticity (i.e., morph ratio) and morphology. We were motivated by the possibility that short-term, nongenetic shifts in plasticity and morphology would offer one explanation for the correlation of *spr-5* selection with morphological divergence in Diplogastridae. Our reasoning was that weakened selection on epigenetic modifiers can

promote phenotypic evolution[22] and even compromised function can be positively selected and adaptive, particularly in enzyme evolution[36]. Using *spr-5* loss-of-function mutants as a powerful proxy of this potential effect, we tested whether multigenerational shifts in mouth morphology were possible by selecting on the epigenetic variation mediated by *spr-5*. For this test, we performed artificial selection on polyphenism phenotypes in highly inbred (i.e., genetically homozygous) but likely epigenetically variable[25–27] *spr-5* lines. Because selection on inbred lines of the wild-type background in which our mutants were made fail to drive any shift in morph bias[12], and given the self-fertilizing reproductive mode and high homozygosity of laboratory strains of *P. pacificus*[37,38], we would interpret any selectable differences over a short time to be most likely upon epimutations rather than standing genetic variation. Likewise, since the mutation rate of this species is typical of many animals (i.e., on the order of 10$^{-9}$)[39], we consider it unlikely that any phenotype could detectably be selected from new mutations over a small number of generations.

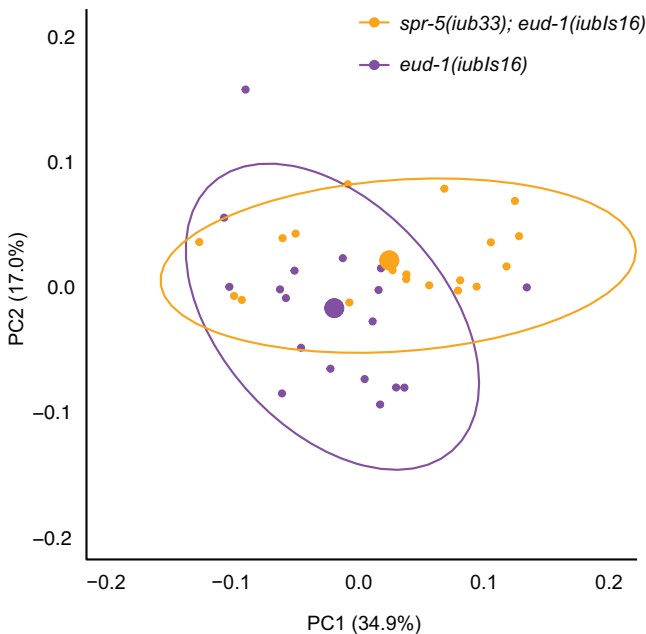

**Fig. 3 | *spr-5* influences morphology in the absence of plasticity.** In a genetic background that fixes development for a single (Eu) morph, *eud-1(iubIs16)*, *spr-5* mutants occupy different regions of morphospace. Source data are provided as a Source Data file.

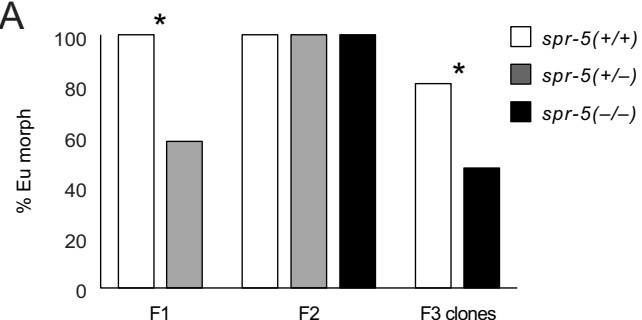

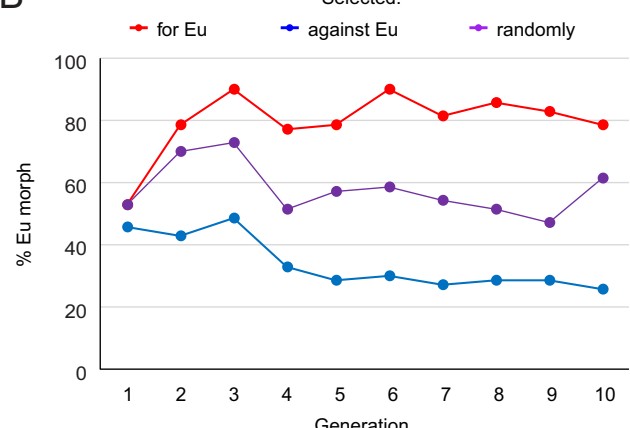

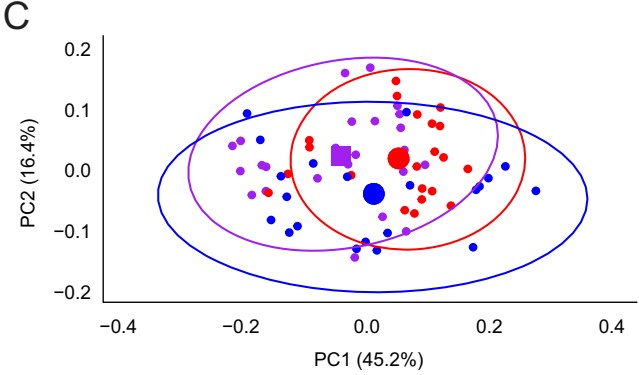

**Fig. 4 | Loss of *spr-5* function enables multigenerational nongenetic inheritance of plastic morphology. A** Outcrossing of *spr-5(iub33)* mutants to the wild-type (PS312) resulted in significantly fewer Eu $F_1$ than did a self-cross of PS312. After returning to the wild-type phenotype in the $F_2$ (all genotypes), $F_3$ individuals cloned from homozygous $F_2$ differed significantly between *spr-5*$^{-/-}$ and *spr-5*$^{+/+}$ lines. $P_0$ females (self-fertile hermaphrodites) in both cross types were marked by an *unc-22* mutation to distinguish $F_1$. Asterisks (*) denote which groups were significantly different following a two-sided Fisher's exact test in the $F_1$ ($P = 3.93 \times 10^{-3}$, Fisher's exact) and two-sided logistic regression in the $F_3$ (estimate = 2.52, $Z = 3.16$; $P = 0.002$). **B** Selection on morphology and for changes in mouth-morph ratio in genetically inbred *spr-5*$^-$ lines leads to a heritable shift in plasticity. **C** Principal components analysis of geometric morphometrics of randomly sampled $G_{10}$ individuals. Source data are provided as a Source Data file.

After 10 generations of selection, the control line, whose morphology was not considered for selection, showed no net change in plasticity as measured by the proportions of categorically defined morphs (% Eu; estimate = -0.041, $Z = 1.21$, $P = 0.23$). However, lines selected either for the Eu morph (estimate = 0.079, $Z = 2.20$, $P = 0.028$) or against it (estimate = -0.128, $Z = 3.93$, $P = 8.48 \times 10^{-5}$) each showed significant shifts in plasticity over the same number of generations (Fig. 4B). These shifts resulted in divergence of morph proportions: whereas lines did not differ in the proportion of Eu to non-Eu (St or intermediate) individuals at $G_1$ (for-Eu estimate relative to random = -0.008, $Z = 0.018$, $P = 0.986$; against-Eu estimate relative to random = -0.214, $Z = 0.71$, $P = 0.478$), they significantly differed at $G_{10}$ (for-Eu estimate relative to random = 0.839, $Z = 2.62$, $P = 0.009$; against-Eu estimate relative to random = -1.531, $Z = 3.25$, $P = 0.001$). Using geometric morphometrics on random samples of individuals from each of the selected lines at $G_{10}$, we also found that the three lines diverged in their morphology, occupying three significantly different regions of morphospace (Fig. 4C; for-Eu *vs.* random: $Z = 2.50$, $P_{adj} = 0.018$; against-Eu *vs.* random: $Z = 2.34$, $P_{adj} = 0.018$; for-Eu *vs.* against-Eu: $Z = 1.82$, $P_{adj} = 0.046$). Additionally, the line selecting against Eu morphology exhibited a significant increase in morphological disparity compared to the other two lines (Fig. 4C; against-Eu *vs.* random: $d = 0.012$, $P_{adj} = 0.012$; against-Eu *vs.* for-Eu: $d = 0.018$, $P_{adj} = 0.003$). In principle, the latter strain's wider range of phenotypes could subsequently serve as substrate for further selection to refine into adaptive forms. Thus, our artificial selection regime resulted in rapid, heritable, and directional change in both plasticity and the resulting morphology.

## Discussion

We have shown that a mediator of nongenetic inheritance, *spr-5*, influences a developmentally plastic trait, exposes variation—presumably in the form of epimutations—in that trait to directional selection, and enables multigenerational shifts in phenotype over few generations. Moreover, we identified this gene from a screen designed to identify candidates from nematode lineages that have both lost plasticity and undergone the rapid evolution of once-plastic forms[6]. Thus, using past events, we have pinpointed a specific factor that can

effect change in polyphenism in the absence of standing genetic variation.

The importance of nongenetic inheritance in evolution has been supported by a surge of examples from the laboratory and in nature [reviewed in[40,41]]. In the laboratory model *C. elegans*, mechanisms for transgenerational inheritance of environmental responses have been described, specifically small RNAs[42] and histone (H3K9me3) methylation[43]. This precedent had suggested that differences in the

more recent nematode innovation studied here, resource polyphenism, might be inherited across generations and likewise through histone modifications. For polyphenism regulation, *P. pacificus* requires H4K5/12 acetylation during a critical window of larval development, such that polyphenism switch-gene (*eud-1*) expression is licensed within a generation[44]. In addition, histone methylation levels (H3K4me2/3) at the same switch-gene have also been linked to changes in gene expression and shifts in mouth form ratio[29]. Here, we have identified a histone modifier that both canalizes the alternative forms of a resource polyphenism (Fig. 2B) and mediates the inter- and multigenerational accumulation of changes to their environmental sensitivity (Fig. 4). Moreover, by drawing on the precision with which epigenetic mechanisms for transgenerational memory are being described, specifically in nematodes[45,46], *spr-5* anchors testable predictions about how developmental plasticity should access such memory.

Another upshot of our findings is that *spr-5* identifies a mechanism joining two axes of variation in polyphenic traits: their environmental sensitivity and ultimate morphology. Discontinuity in plasticity suggests that the sensitivity and the ultimate trait mean should be free to evolve independently. In this case, these two aspects of plasticity should involve alternative sets of genes[47,48], a premise that has been supported by empirical studies, for example, of feeding morphology in cichlids[49] and resource polyphenism in spadefoot toads[50]. Testing this idea at a molecular level, we show that sensitivity and trait values can also be influenced by the same molecular processes. Previously, all described regulators and effectors of the *Pristionchus* mouth polyphenism had reported influence on just one axis of variation. Whereas some affect the ratio under which a stereotypical morph is produced, through effects on either sensory perception or endogenous signalling[16,17,51,52], mutations in others cause morph-constitutive phenotypes that assume aberrant but nevertheless consistent forms[15,53]. Presumably through its genome-wide effects as a chromatin modifier, *spr-5* shows mutant phenotypes both in plasticity and in morphological trait values.

Given that the evolution of both plastic and assimilated forms in Diplogastridae has been quantitative, rather than the simple fixation of alternative forms throughout the family[6], morphological change associated with plasticity and its loss has likely been complex, involving multiple genetic modifiers rather than an occasional large effect on the switch pathway itself. Because *spr-5* controls exposure of morphological variation, gene expression influenced by *spr-5* will help to identify such modifiers, which should include the ultimate targets of the polyphenism. Specifically, differences in the H3K4me1/2 landscape among species, strains, and environments would highlight these targets. It is also possible that *spr-5* itself has a role in this process, since it influences both what variants are visible to selection and how often they are done so. Comparative analyses combined with functional analyses of polyphenism targets, as we have performed here for *spr-5*, will allow tests of this principle. Moreover, because *spr-5* plays a role in DNA damage repair[54], modified function of this gene might result, over more generations than we examine here, in de novo mutations directly. In this way, it is possible that the multigenerational maintenance of a selected phenotype through nongenetic inheritance might facilitate both the sorting of standing genetic variation and the accumulation of new variation. Re-sequencing of selected *spr-5* lines would formally test whether this is a possibility in real time in the laboratory.

Finally, a mechanistic understanding of modifiers of plastic morphologies, such as *spr-5*, will inform whether and how divergence within morphs ultimately influences the birth of new morphs, cases of which are well documented for resource polyphenism[55]. For instance, if epigenetic regulators promote and stabilize, even over a few generations[41], novel morphologies that have altered resource-use capabilities, then competition and selection could act as wedge to exaggerate induced differences and push forms into new regions of niche and phenotypic space[56]. Indeed, in at least two cases in Diplogastridae, new morphotypes have been apparent exaggerations of previously existing ones, such as the novel feeding-types in several fig-associated *Pristionchus* species[57] and the "teratostomatous" morph of *Allodiplogaster sudhausi*[58]. Knowledge of polyphenism targets whose expression is mediated by *spr-5* will allow tests of their role in the rapidly diverging mouth morphologies within and among lineages of Diplogastridae.

In conclusion, our findings highlight a mediator of epigenetic inheritance, specifically an H3K4 di/monodemethylase, as a facilitator of change in traits as they traverse the continuum of greater to lesser environmental sensitivity. Characterization of this process gives a mechanistic basis to longstanding theory and makes a guidepost for exploring the generalizability of genetic assimilation and its molecular underpinnings in diverse systems.

## Methods

### Ablation of polyphenism switch genes in *P. fissidentatus*

Gene editing by CRISPR/Cas9 was used to knock out orthologues of *eud-1* and *seud-1/sult-1* in *P. fissidentatus* (strain RS5133). Target sequences for *eud-1* and *seud-1/sult-1* were chosen to be in coding sequences that, when deleted in *P. pacificus*, result in loss-of-function mutations. Protospacer adjacent motif (PAM) sites and corresponding crRNAs were then identified within the exons containing these sequences in *P. fissidentatus*. tracrRNA and designed crRNAs were synthesized by Integrated DNA Technologies (IDT). gRNAs were produced following the manufacturer's protocol and combined with Cas9 to generate a ribonucleoprotein (RNP) complex. The final injection mix consisted of 0.43 μl of the RNP complex in 4.57 μl of TE buffer, together with a plasmid (50 ng/μl) containing a *P. pacificus* codon-optimized *egl-20p::TurboRFP::rpl-23^{UTR}* construct as a co-injection marker for identifying successful injections among $F_1$ individuals[59]. All marker-reporting $F_1$ and their siblings were isolated and allowed to self-cross. After eggs ($F_2$) were laid, $F_1$ were genotyped using a heteroduplex mobility assay (HMA)[60]. Offspring of $F_1$ determined to be heterozygous for a CRISPR/Cas9-mediated mutations were isolated and allowed to self-cross, upon which these $F_2$ mothers were genotyped by HMA and homozygous mutant lines analysed by Sanger sequencing. Sequences for guide RNAs and primers are given in Supplementary Table 2.

### Phenotype scoring of *P. fissidentatus* mutants

To confirm that the phenotypic effects of *eud-1* and *seud-1/sult-1* are the same in *P. fissidentatus* as in *P. pacificus*, the mouth morphs of 60 individuals from each of five plates ($n = 300$) were scored for the wild-type strain (RS5133), *Pfi-eud-1(iub27)*, and *Pfi-seud-1(iub22)* under standard laboratory conditions, namely: on 6-cm plates with nematode growth medium agar, fed a lawn of *Escherichia coli* OP50 grown in 300 μl L-broth, and maintained at 20–25 °C. Additionally, the former two strains were starved in attempt to induce higher proportions of the Eu morph[12], to test the penetrance of the *Pfi-eud-1(iub27)* mutant phenotype. Starvation conditions were achieved by allowing a second generation of nematodes to emerge and develop on plates on which all bacterial food was exhausted by the time that generation reached adulthood (*i.e.*, 10 days after seeding plates with $G_0$ mothers). All phenotypes were scored using a Zeiss Axio Imager A1 compound microscope. Phenotypes were assigned according to previous descriptions of the mouth morphology[19]. Briefly, mouth width, the shape of the dorsal tooth, and the presence of a right subventral tooth diagnose Eu and St morphologies. Whereas Eu individuals have a mouth wider than deep, a heavily sclerotized, claw-like dorsal tooth, and a well-developed right subventral tooth, St individuals have a mouth narrower than deep, a weakly sclerotized, spade-like dorsal tooth, and no subventral tooth. Differences in morph ratio were analysed using a mixed model logistic regression with treatment as a fixed

effect and plate as a random effect, as implemented in the R (version 4.0.1) package lme4[61].

## RNA extraction and sequencing of *P. fissidentatus* mutants

Once mutant lines of *P. fissidentatus* were confirmed to be fixed for alternative morphs of the mouth polyphenism, as in *P. pacificus*, we performed RNA-seq to identify morph-associated genes[62,63]. For each mutant line, nematodes were rinsed from five 100 mm-diameter plates containing mixed-stage cultures into M9 buffer in a 50-ml collection tube. 40 µl ampicillin (50 µg/µl) and 40 µl chloramphenicol (25 µg/µl) were added to a 40 ml solution of 0.9% NaCl containing the collected nematodes. Nematodes were gently shaken for 2 h at room temperature, spun at 1300 g to create a pellet, stored in 1 ml TRIzol (Invitrogen cat. no. 15596026), and frozen at −80 °C until extraction, which was performed as described previously[62] using the manufacturer's protocol for the Zymo Direct-zol RNA miniprep kit (Zymo Research cat. no. R2052). This procedure was performed in duplicate for each mutant line. Libraries from total RNA extracts were prepared using the Illumina TruSeq Stranded mRNA HT Library kit and were sequenced as paired-end, 75-bp reads on the Illumina NextSeq platform, producing ~40 million read-pairs per library. Reads trimmed of adapters were quality-filtered using a cut-off threshold for an average base quality score at 20 over a window of 3 bases. Reads shorter than 20 bases were excluded, resulting in ~90% of reads with both mates passing quality filters. The filtered reads were then mapped to the *P. fissidentatus* genome using STAR v. 2.7.3a[64] with ~91% of reads aligning to the reference sequence. The featureCounts tool v. 2.0.0[65] was used to quantify expression by counting reads that mapped to exon regions of annotated genes.

## Inference of ancestral, environmentally sensitive genes in *Pristionchus*

Genes with morph-biased expression in *P. fissidentatus* were identified by comparing the transcriptomes of *Pfi-eud-1* and *Pfi-seud-1* mutants. Transcriptomes were sequenced for mixed-stage animals, so that expression throughout development would be captured, as done previously for switch-gene mutants of *P. pacificus*[62]. This was done using the combined results of DESeq2[66] and weighted gene co-expression network analysis[67], as used to identify polyphenism-biased genes in *P. pacificus*[62,63]. If the datasets from these two analyses disagreed on a given gene's morph-bias, the notation based on DESeq2 was selected. Cataloguing of the output from these two datasets generated a list of ~8000 genes with morph-biased expression in *P. fissidentatus*. This list was then compared by BLAST to genes showing differential expression[62], co-expression[63], or both in *P. pacificus* (~3000 morph-biased genes). Because not all genes were reciprocally best BLAST hits between species, all unique, orthologous pairs were retained in the final data, such as to maintain as many candidate polyphenism genes as possible. BLAST was used to identify the most up-to-date annotation for *P. pacificus* (prisitonchus.org), and then morph association in both species was compared. We retained those genes that exhibited concordance in morph-biased expression as our core set of polyphenism-associated genes.

## Comparative genomic tests between polyphenic and monomorphic *Pristionchus* species

Transcriptomes for *Pristionchus* were compared to identify genes uniquely evolving in *P. bucculentus* and *P. elegans*, which underwent genetic assimilation of a single morph during evolution. This analysis used previously published transcriptomes and orthogroup assignments for 39 *Pristionchus* species plus two outgroup species, *Micoletzkya japonica* and *Parapristionchus giblindavisi*[21]. For each orthogroup, branches belonging to *P. bucculentus* and *P. elegans* were tested for episodic, diversifying (positive) selection. This test was performed using ~7000 orthogroups that were required to contain both *P. bucculentus* and *P. elegans*, at least one outgroup (non-

*Pristionchus*) species, and at least two polyphenic *Pristionchus* species. Because some transcriptomes were only available in the reverse complement direction for each orthogroup, all sequences were placed in the same orientation (5′ to 3′) by adding the *P. pacificus* (or if it was not present for a given orthogroup, *Micoletzkya japonica* or *Parapristionchus giblindavisi*) sequence (obtained from genome sequencing and annotation done previously) to the combined FASTA file and using this sequence as a calibrator to orient the other sequences with MAFFT v. 7.471[68]. These calibrator sequences were then removed from downstream analyses. Sequences were then codon-aligned using the *alignSequences* function in MACSE v. 2.03[69], had gap-rich portions of the amino acid alignments removed using the -partial flag option in Divvier v. 1.01[70], and then refined using these amino-acid alignments with MACSE, using the reportMaskAA2NT function. These alignments served as input for the program aBSREL in the package HyPhy v. 2.5.21[71] to test for branch-specific, episodic diversifying selection. Gene trees were inferred from the final codon-aligned sequences, under a GTR model and a gamma distribution with 25 rate categories, in RAxML v. 8.2.11[72].

In addition, a test for a weakening of selection on orthogroups in assimilated species was performed. For this analysis, we set all gene copies of a given assimilated species as the foreground being tested for relaxed selection relative to the rest of the tree (set as reference). This conservative approach limits our findings to only those orthogroups, rather than individual genes, in which an assimilated species is experiencing a weakened strength of selection. Individual genes or, in the case of relaxed selection, orthogroups were classified as polyphenism, non-polyphenism, and, for the polyphenism genes, Eu or St based on the classification of their best match to their orthologues in *P. pacificus*. Tests for weaker intensity of selection were performed with the program RELAX[73], as implemented in HyPhy. RELAX evaluates the relative intensity of selection acting on a set of focal branches compared to other branches and considers both the intensity of purifying and diversifying selection. This intensity is indicative of selection pushing ω values toward 1 in the focal branch(es) compared to the reference branches. Because the test subjects of RELAX are different from that of aBSREL—whereas the former program detects whether intensity of selection has weakened (for both purifying and diversifying selection) across a set of foreground branches relative to the rest of the tree, the latter determines if some proportion of sites along a branch have evidence for diversifying selection—it is possible for the results of both types of analysis to identify the same genes. That is, relaxed purifying selection and episodic diversifying selection are both consistent with positive shifts in ω. Therefore, the gene lists from both analyses were intersected to identify increased diversity, as indicated by both signals of selection, in genes in the assimilated lineages.

## Comparative genomic tests between independent assimilation events

To test the generalizability of our findings on genetic assimilation in *Pristionchus*, comparative genomic tests for selection between species from an entire clade that experienced genetic assimilation and its sister clade that ancestrally had the mouth polyphenism[6] were also performed. This analysis was performed separately from that of *Pristionchus* and close outgroups so that the number of orthogroups retained for each of the two analyses was maximized. Here, Ortho-Finder v. 2.5.1[74] was used to identify orthogroups for 14 diplogastrid nematode species for which reference genomes were available, upon which alignments were generated and gene trees were inferred, as described above. Only orthogroups with (i) the outgroups *Allodiplogaster sudhausi* or *Koerneria luziae*, (ii) at least two polyphenic species from different genera (*Micoletzkya*, *Parapristionchus*, *Pristionchus*), and (ii) both species representing this assimilation event (*Diplogasteroides magnus* and *Levipalatum texanum*) were included. Results from aBSREL and RELAX were then compared to those from

assimilation of *P. bucculentus* and *P. elegans*. Significance values from these tests were adjusted across all species using false discovery rate of 0.05. Among polyphenism-biased genes showing diversifying and relaxed selection strength was a gene annotated in *P. pacificus* as ppa_stranded_DN16305_c0_g1_i1 (El Paco v. 3 genome assembly, pristionchus.org), the single orthologue of *C. elegans spr-5*, as confirmed by reciprocal best BLASTp.

## Inference of effects of amino-acid changes in species with assimilated forms

To gain insight on what functional effects that *spr-5* mutations might have had in species with assimilated forms, ancestral state reconstruction was combined with protein modelling. Prior to ancestral state reconstruction, the *spr-5* coding-sequence alignment was manually curated, and the gene-tree inference refined. Sequences aligned as described above were further trimmed of ambiguously aligned sites at the 5′ terminus using MEGA v. 11[75]. The manually edited alignment was used to infer the tree in RAxML, under a GTR model and a gamma distribution with 25 rate categories, specifying *Micoletzkya japonica* as outgroup. This tree had a topology identical with that inferred prior to our selection analyses and was used for downstream analyses, described below.

With a more conservative alignment and corresponding gene-tree, an ad hoc test for diversifying selection at individual sites across *spr-5* was performed. For this test, the package codeML, as implemented in PAMLX v. 1.3.1[76], was used to perform Bayes Empirical Bayes (BEB) analysis[77] to get Bayesian estimates of the likelihood of each site being under diversifying selection in *P. bucculentus* or *P. elegans* relative to the rest of the tree. Those sites with an $\omega > 1$ and a BEB probability of >0.9 were retained as candidates for diversifying selection.

Given this set of amino acids reporting selection, their potential influence on protein function in *P. bucculentus* and *P. elegans* was explored. First, the codon alignment was converted to an amino acid alignment with the *translateNT2AA* function in MACSE. Next, ModelTest-NG[78] was used to infer the best-fitting model of protein evolution among these sequences across the inferred gene tree according to both Akaike and Bayesian information criteria, which gave the LG model as best fit. Using RAxML, marginal ancestral states on the gene tree were inferred, with the analysis invoking the PROTCAT model, an LG substitution matrix, and a gamma distribution with 25 rate categories. The reconstructed sequence for the node subtending *P. bucculentus* and *P. elegans* was extracted and aligned to the two species' sequences. Sites in which each species diverged from the inferred ancestor were then found. To predict functional changes caused by these variants, the inferred ancestral sequence was used as input into the Phyre2 protein modelling and recognition tool[79]. Upon identifying the top sequence match as a reference, Phyre2 Investigator was run to identify various features of the protein such as predicted secondary structure, catalytic, pocket, and protein-interface sites, and site mutational sensitivity. Using this information, candidate sites reporting diversifying selection were compared with other diverging sites to determine any differences in the putative functional consequences between the two and in locations across the four main protein domains.

## Functional tests of *spr-5* as a polyphenism regulator

The potential role of *spr-5* in the mouth polyphenism of *P. pacificus* was assessed by creating a loss-of-function mutant of the gene in that species. Using CRISPR/Cas9 with homology-directed repair, a premature stop codon, together with a PstI restriction enzyme cut-site for genotyping, was inserted into the gene to ablate its function. Sequences of repair template and detection primers given in Supplementary Table 3, and the mutant allele is shown in Supplementary Fig. 1. Heterozygous mutants were identified by an HMA, and homozygous mutants were identified by PCR and PstI digestion, followed by confirmation by Sanger sequencing. Two tests were then done to determine the role of this gene in mouth form regulation.

First, after ten generations of maintaining homozygous mutant lines, the mouth morphs of these mutants ($n = 302$) and the wild-type, reference strain PS312 ($n = 300$) were scored, using a Zeiss Axio Imager A1 microscope, according to previously established phenotype-scoring criteria[80], which are the same as those described above for *P. fissidentatus*. However, during this phenotype screen, it was observed that some mutant nematodes were apparently intermediate in their mouth morphologies, which was initially confirmed by the blind scoring of such phenotypes. For comparison to the wild-type strain PS312, these intermediates were categorized first as St then as Eu to determine whether the scoring of these individuals influenced any potential difference in morph ratios from the wild-type. Differences in morph ratio were analysed using logistic regression as described above. Whiskers on bar charts (Fig. 2C) represent the standard error of the mean.

Second, geometric morphometrics[6,20,81] was used to quantify and compare mouth-morphology differences among strains. Using 22 landmarks (Supplementary Table 4), the mouth morphology of *spr-5(iub33)* mutants ($n = 72$), of the Eu morph of the highly Eu-biased wild-type reference strain PS312 ($n = 20$), of the St morph of the St-biased strain RS5200B ($n = 20$), and of both morphs of strain RS5205 ($n = 20$ for each), which has relatively even numbers of both morphs in laboratory culture, was characterized. Because *spr-5(iub33)* mutants showed some intermediate morphologies, individuals were not assigned to morphs a priori but, instead, cluster-based modelling was used to distinguish groups in morphospace[82]. Modelling identified two groups that corresponded to Eu-like and St-like individuals. Thus assigned to morphs, the distance in morphospace between these groups was measured and compared to that between naturally occurring Eu and St morphs, using phenotypic trajectory analysis in the R package RRPP[83].

## Tests of the influence of *spr-5* on plasticity under morph-induction cues

To assay for effects of *spr-5* on plasticity, we reared individuals under conditions that differentially promote production of the alternative morphs in wild-type strains. Specifically, *spr-5(iub33)* mutants and the reference (PS312) strain were reared in liquid culture (M9 buffer) and on nematode growth medium agar plates, which promote the St and Eu morphology, respectively, in the reference strain[14]. Eggs were harvested for both strains using standard *C. elegans* NaOH/bleaching procedures[84] and resuspended in M9 buffer. The number of eggs per μl was estimated by counting the number of eggs in a 2-μl droplet and then the appropriate volume for ~500 eggs was added to five replicates per treatment. After five days, the mouth morph of 30 adults per replicate, for a total of 150 individuals per genotype-treatment combination, was evaluated as above. Due to a spilled beaker, one liquid culture treatment failed for *spr-5(iub33)*, so the final sample size for this treatment was 120 individuals.

We also measured differences between the wild-type and *spr-5* mutants in their response to starvation cues. For this assay, we used the *P. pacificus* strain RS5200B because it is normally St-biased in standard laboratory culture, offering more power to detect differences in induction of the Eu morph. We compared the starvation response to nematodes harbouring a loss-of-function mutation, which was generated in the RS5200B strain (allele *iub44*) by CRISPR/Cas9 as described above. The mutant strain was homozygous for *spr-5(iub44)* for three generations, such that the $F_4$ from microinjected $P_0$ were used as to start of the starvation induction assay, as described for *P. fissidentatus* above. Thirty adult hermaphrodites from each of 10 replicates ($n = 300$ individuals per genotype-treatment combination) were phenotyped, also as above.

For both plasticity assays, differences in morph ratio and evaluation of a genotype-by-environment effect were determined using

logistic regression as above with one difference: here, we compared a model with a treatment-by-genotype interaction term to one without the interaction and evaluated, individually, the terms of the interaction model for significance. If there was evidence of a significant interaction, we then performed ad hoc analyses for each genotype to evaluate the magnitude of treatment effects. For these ad hoc tests, we applied a Bonferroni correction to treatment associated $p$-values before assessing the significance of treatment on each genotype. Boxplots (Fig. 2A, B) are based on the percent of individuals exhibiting the Eu morph per replicate and points surrounding each plot are the percentages for each replicate.

### Assay for *spr-5* influence in the absence of plasticity

To determine if *spr-5* can influence mouth morphology even when the development of a single mouth form is canalized, we first generated a strain of *spr-5* modified by CRISPR/Cas9 to carry a loss-of-function allele of *unc-22*, which results in a recessive morphological marker. *unc-22* mutants were generated as random knockouts, as described for *P. fissidentatus* mutants above. We then crossed this strain with the constitutively Eu strain RS2651 [*eud-1(iubIs16)*] to ultimately generate a *spr-5* mutant line carrying this transgene. After confirming homozygosity of the *spr-5* mutation and, by expression of the integrated RFP reporter, the presence of *eud-1* overexpression construct, we used geometric morphometrics as above to determine if the double-mutant morphology differed from the that of the single mutant. Twenty adults were measured for each genotype.

### Tests of *spr-5* as a mediator of nongenetic inheritance of plasticity

To determine the intergenerational effects of *spr-5* loss and associated changes in gene regulation, crosses were performed between *spr-5(iub33)* mutants and the wild-type (PS312). Individuals were characterized as Eu or non-Eu using a Zeiss Discovery.V20 stereomicroscope and differences in phenotype between this test cross ($n = 12$) and a control cross ($n = 20$) using only wild-type $P_0$ were assessed using Fisher's exact test. To distinguish $F_1$ of $P_0$ hermaphrodites, which are self-fertile, the wild-type (PS312) and *spr-5(iub33)* strains were both modified by CRISPR/Cas9 to carry loss-of-function alleles of *unc-22* as described above. Following crosses of morphologically marked $P_0$, $F_1$ were allowed to lay eggs and then the $F_2$ phenotyped ($n = 360$). Randomly chosen $F_2$ were then isolated, allowed to lay eggs, and genotyped. $F_3$ clones of homozygous $F_2$ mutants ($n = 270$) and wild-type individuals ($n = 330$) were then phenotyped and their morph ratios compared using logistic regression.

### Tests of product inhibition of SPR-5 on the polyphenism threshold

The mechanism of *spr-5*-mediated nongenetic inheritance was investigated by exposing wild-type (PS312) individuals to the compound Bizine (Cayman Chemical item no. 19705), which is an inhibitor of the human SPR-5 orthologue LSD1/KDM1A[85]. Three replicate lines were exposed to 15 µl of Bizine (5 µg/µl in DMSO) + 85 µl DMSO (16.8 µM final concentration in a 15-ml agar plate) and three control lines received 100 µl DMSO. Four J4 (pre-adult) individuals were put on each plate and their offspring were phenotyped ($n = 50$ per plate) using a Zeiss Axio Imager A1 microscope Because the effects of SPR-5 inhibition accumulate over generations in *C. elegans* examples[25–27], four individuals were randomly picked to produce a second generation, so that lines of mutant nematodes could be scored for their phenotype under the same conditions over several generations. Specifically, prior to transferring nematodes each generation, the plates were either treated with Bizine or DMSO as described above. This experiment was performed over eight generations. The numbers of Eu and non-Eu

individuals were analysed by logistic regression as above and the number of intermediates emerging from both treatments was compared with Fisher's exact test.

### Artificial selection to test the role of *spr-5* in plasticity evolution

Artificial selection was performed on homozygous mutant lines drawn from a common stock. Specifically, to increase and decrease the Eu:non-Eu ratio, in each generation one culture plate was selected from among five plates per line that showed the highest and lowest Eu:non-Eu ratios, respectively, to propagate the next generation. Mouth-morph ratio, while measured, was ignored as a criterion for selection during random propagation of the control line. Artificial selection was thus carried out for 10 generations. For each line, a logistic regression model was fit, with plate as a random effect, to evaluate if the morph ratios changed over generations ($n_{random} = 1293$; $n_{for-Eu} = 1828$; $n_{against-Eu} = 700$). In addition, at generations 1 and 10, morph ratios among lines were compared ($n_{G1} = 313$; $n_{G10} = 288$). Finally, adults were randomly sampled at generation 10 for geometric morphometric analysis using the landmarks described above ($n_{random} = 25$; $n_{for-Eu} = 23$; $n_{against-Eu} = 22$). Morphologies among lines were compared using the *procD.lm* and *pairwise* functions in the R packages geomorph and RRPP, respectively. The function *morphol.disparity* from the geomorph package was used to compare variation in morphology among selected lines.

### Reporting summary

Further information on research design is available in the Nature Portfolio Reporting Summary linked to this article.

## Data availability

Transcriptomes were accessed from pristionchus.org (http://pristionchus.org/download/Roedelsperger_et_al_2018_pristionchus_transcriptome_assemblies.tgz). Genomes were accessed from pristionchus.org (http://pristionchus.org/download/diplogastrid_annotation_ppcac_v1.tgz, http://pristionchus.org/download/El_Paco_genome.tgz) and from NCBI (https://www.ncbi.nlm.nih.gov/nuccore?term=PRJNA655932). Newly generated RNA sequencing reads have been deposited in the NCBI Sequence Read Archive (https://www.ncbi.nlm.nih.gov/bioproject/PRJNA919017). Lists of polyphenism-associated genes, genes under selection, and phenotypic data are available in the Dryad Digital Repository[86] (https://doi.org/10.5061/dryad.98sf7m0nm). All other data needed to evaluate the conclusions in the paper are present in the paper, the Supplementary Information, or the Source Data file. Source data are provided with this paper.

## Code availability

R code for analysing data is available in the Dryad Digital Repository[86] (https://doi.org/10.5061/dryad.98sf7m0nm).

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

## Acknowledgements

We thank the Indiana University Centre for Genomics and Bioinformatics for performing RNA sequencing of *P. fissidentatus* mutant lines. This work was funded by the United States National Science Foundation (PRFB-2109325 to N.A.L. and grants IOS-1911688 and IOS-2229383 to E.J.R.).

## Author contributions

N.A.L. and E.J.R. conceptualized and designed the study, interpreted results, and wrote the manuscript. N.A.L. performed the experiments and analysed the data with input from E.J.R.

## Competing interests

The authors declare no competing interests.
