## [Peer Review File · Nature Communications]

A histone demethylase links the loss of plasticity to nongenetic inheritance and morphological changeREVIEWER COMMENTS

Reviewer #1 (Remarks to the Author):

Key Results

This manuscript describes an integrative and complementary set of approaches to address the roles of developmental plasticity and epigenetic inheritance in evolutionary processes. Using a clade of nematode worms that exhibit an ancestral trophic polyphenism, and which has undergone repeated episodes of genetic assimilation, the authors come to the following main conclusions:

1. The polyphenic switch mechanisms characterized in the model *P. pacificus* are general to the genus, and thus can be manipulated to identify morph-biased genes ancestral to the entire *Pristionchus* clade.
2. Independent episodes of genetic assimilation in *Pristionchus* have involved positive selection on a similar set of morph-biased genes. One of these genes, *spr-5*, is a known regulator of trans-generational, epigenetic inheritance, suggesting that evolutionary changes to this epigenetic modifier may be an important mechanism of genetic assimilation.
3. By ablating *spr-5* and blocking the function of its protein product (which demethylates histone H3K4), the genetically assimilated phenotypes were recapitulated in the lab, and mutants demonstrated increased phenotypic variance. The latter conclusion suggests that mutations to *spr-5* in nature may act as a capacitor for subsequent evolutionary change by revealing new phenotypic targets to selection.
4. The polyphenic effects of *spr-5* carried through one generation, indicating that *spr-5* can mediate plasticity transgenerationally.
5. In 10 generations of artificial selection on epigenetic variation, the authors selected for both increased and decreased production of the Eu morph. Thus, they demonstrated that selection on epigenetic variation can lead to the evolutionary shifts in plastic responses (i.e., genetic accommodation broadly, genetic assimilation specifically).

Validity

Based upon my expertise (see below for more information), I found the data interpretations to be valid and compelling. Not only were the results of individual approaches valid, but the fact that multiple, complementary approaches reinforced a variety of conclusions was especially convincing. For example, the potential role of *spr-5* in genetic assimilation was established by ablating the gene, blocking its product, and through artificial selection.

Significance

For more than a century, and increasingly in the past few decades, a variety of theories have been proposed regarding whether and how non-genetic phenomena such as phenotypic plasticity and trans-generational epigenetic modifications might constrain, promote, or otherwise modify the trajectory of adaptive evolution. Accumulating empirical evidence has supported some of these ideas, which are the subject of a major debate within Evolutionary Biology (i.e., whether the Modern Synthesis should be revised as the Extended Evolutionary Synthesis to account for these less traditional mechanisms). However, few, if any, papers that I can recall address these issues in such a thorough and detailed way.

This manuscript describes a series of complementary studies and experiments through which several of these important threads are woven together with convincing empirical support. These include the roles of genetic assimilation of previously plastic responses, trans-generational inheritance of epigenetic modifications, and the potential for non-genetic variation to act as evolutionary capacitor. In my view, this is an exceptional paper that is likely to become a classic study of the mechanistic basis genetic assimilation, epigenetic inheritance, and their evolutionary consequences. The integration of bioinformatics, comparative transcriptomics, direct genetic manipulation, morphometrics, and experimental evolution paint an extraordinarily detailed and compelling picture of how genetic assimilation has likely repeatedly occurred in this genus of

nematodes, and how the mechanism of genetic assimilation might act as a capacitor for further evolutionary change.

Data and Methodology

I judged the methods to be elegant, logical, and mutually supportive. The data appear valid and reliable, insofar as my expertise allows me to fully evaluate them. While I do not directly engage in molecular approaches in my own lab, I collaborate with biologists who do, and am confident in my ability to evaluate the results of molecular studies with respect to questions of mechanisms of evolutionary and trans-generational inheritance. I cannot comment on the validity of the molecular methods, but the interpretation of their results seems sound. I do have expertise in morphometric methods, and I do find these approaches and results to be valid.

Analytical approach

The analytical approaches are valid.

Suggested Improvements

I have only minor suggested improvements:

Fig.1A—Is this phylogeny reproduced from reference 7 (Fig. 2 therein), or generated as part of the comparative genomic tests for genetic assimilation? I suspect the phylogeny came from reference 7, which would then be used to identify comparisons of interest for the comparative genomic analysis described in the present study. If that's the case, then the figure legend should cite reference 7.

296: change to "were scored" instead of "where scored"

298: I interpret "under standard laboratory conditions" to indicate that individuals from these three strains were raised in conditions that typically generate low proportions of the Eu morph. Can the authors cite a reference for their "standard laboratory conditions," or otherwise describe them briefly? Given that the authors are investigating a polyphenism, the details of the environmental conditions are important.

306: What R version, and which R package(s) were used?

Clarity and Context

The text is well-written and the figures and tables are clear, thorough, and should be accessible to a wide audience. This clarity is especially important given that this manuscript uses a variety of integrative approaches, so any given reader might not be expert in every aspect of the paper.

References

The references cited are appropriate in that they support the broad theoretical basis of the manuscript, as well as the methodological approaches.

Reviewer Expertise

My expertise is in the role of plasticity in evolutionary adaptation, particularly with respect to the evolution of morphology. I collaborate on projects involving bioinformatic and genomic approaches, so I feel confident in my ability to evaluate the rationale and results of all aspects of the manuscript. However, I do not myself use molecular techniques, so I am not an expert in the specific methodologies by which these results are generated.

Reviewer #2 (Remarks to the Author):

The article 'A regulator of nongenetic inheritance mediates the evolution and loss of plasticity' by Levis and Ragsdale reports the finding of a histone tail modifying enzyme (*spr-5*) that is associated with morph-biased gene expression between two types of feeding morphology and that has been under positive selection in two lineages that are fixed for one type of feeding morphology. Since

spr-5 is also a known and demonstrated mediator of epigenetic inheritance, the authors interpret spr-5 as an "epigenetic bridge" that enables (transgenerational) plasticity and that is the target of selection during genetic assimilation.

The manuscript is addressing an interesting idea and presents some suggestive results that are broadly consistent with these ideas. However, the presented data is also consistent with a number of alternative interpretations (see below) that are not sufficiently acknowledged, and conclusion do not rigorously follow from the results. Several additional lines of evidence would be required to make this a truly convincing piece of work. For example, more information on the function of spr-5 in the lineages that are fixed for one morph would be desirable and strengthen the case. I have several major criticisms about the rationale presented in the study and the soundness of the interpretations.

Given the evidence presented, the authors make quite far-reaching statements. For example, in lines 51-53: the authors have established two lines with knock-outs in two switch genes that cause the nematodes to constitutively express one of the two morphs. Using differential gene expression between the two lines, they identify genes with a morph-biased gene expression. Having done this for two *Pristionchus* species and retaining the overlap of these morph-biased genes, they claim that they identified a 'core set of environmentally sensitive genes ancestral to *Pristionchus*'. Calling these genes 'environmentally sensitive' is quite a stretch since they really are differentially expressed between two mutant lines, and it is far from given that the transcriptomic signature of a CRISPR-induced morph is the same as that of an environmentally-induced morph. As a side, the reader is never told what environmental cue naturally induces the morphs. Why are not naturally-occurring morphs used to identify a set of environmentally sensitive genes? In addition, using two species with unclear phylogenetic positions (after all the reader is told, those could be very closely related, highly derived sister species) is not strong evidence for the set of 'genes inferred to be ancestral to the entire genus' (line 53).

-selection-based analysis (lines 54-70): This section would greatly benefit from more details and it's very hard to interpret how convincing the data actually is. How many genes were identified as positively selected in the two species that allegedly recently lost plasticity? How many genes in the two species that lost plasticity longer ago? Also, no evidence is provided that the two events of plasticity loss are actually on different time scales since the phylogeny in Fig. 1 is not dated. Related to this, I am not convinced that looking for genes that have experienced positive selection in the past is the right type of analysis here. Why would a gene with a morph-biased expression be under positive selection in a lineage constitutively expressing one of the morphs? Given the direction of morph-biased expression, a really important gene might as well have become pseudogenized or lost its function in the lineage that fixed the morph in which it's not expressed. At the very least, the analysis should be performed directional, meaning to look for overlap of type-A-biased genes that are under positive selection in species only exhibiting type-A (but not type-B) morphologies. At present, this analysis seems poorly thought-through, superficially conducted and insufficiently described.

-the authors selected spr-5 among the 19 genes that were identified as having a morph-biased gene expression in *Pristionchus* and having experienced positive selection in at least one species of the two lineages that lost plasticity. What about the other 18 genes? Given their equally strong association with the morphs as spr-5, it is well conceivable that their ablation might also have caused differences in the morph-expression, similar to spr-5. What would this mean for the interpretation of the role of spr-5?

-arguably using a loss of spr-5 function to test whether this modification could allow directional selection upon new morphologies as observed in species with fixed morphs is not ideal. Ablating gene function is rather the opposite of mimicking the effect of positively selected alleles, and therefore I am not sure what the biological relevance of a spr-5-negative organism in the context of this study is. Since the logic here is reverse to what I would have thought is relevant to test, I do not follow the interpretation of the authors, and the far-reaching conclusion that they draw based on this experiment.

-minor question: are the 838 genes identified as differentially expressed in both species

concordantly differentially expressed? I.e., are the 838 genes up- and down-regulated in the same morphs between the two species? If not, those genes should be filtered to retain only those with consistent expression changes since the discordant ones are probably false-positives.

Overall, I find the presented ideas interesting and the presented results suggestive, but there are several important gaps in the logic that unfortunately mean that the conclusions are not fully supported by the presented evidence.

RESPONSE TO REVIEWERS' COMMENTS

Reviewer #1 (Remarks to the Author):

Key Results

This manuscript describes an integrative and complementary set of approaches to address the roles of developmental plasticity and epigenetic inheritance in evolutionary processes. Using a clade of nematode worms that exhibit an ancestral trophic polyphenism, and which has undergone repeated episodes of genetic assimilation, the authors come to the following main conclusions:

*1. The polyphenic switch mechanisms characterized in the model *P. pacificus* are general to the genus, and thus can be manipulated to identify morph-biased genes ancestral to the entire *Pristionchus* clade.*

*2. Independent episodes of genetic assimilation in *Pristionchus* have involved positive selection on a similar set of morph-biased genes. One of these genes, *spr-5*, is a known regulator of trans-generational, epigenetic inheritance, suggesting that evolutionary changes to this epigenetic modifier may be an important mechanism of genetic assimilation.*

*3. By ablating *spr-5* and blocking the function of its protein product (which demethylates histone H3K4), the genetically assimilated phenotypes were recapitulated in the lab, and mutants demonstrated increased phenotypic variance. The latter conclusion suggests that mutations to *spr-5* in nature may act as a capacitor for subsequent evolutionary change by revealing new phenotypic targets to selection.*

*4. The polyphenic effects of *spr-5* carried through one generation, indicating that *spr-5* can mediate plasticity transgenerationally.*

*5. In 10 generations of artificial selection on epigenetic variation, the authors selected for both increased and decreased production of the *Eu* morph. Thus, they demonstrated that selection on epigenetic variation can lead to the evolutionary shifts in plastic responses (i.e., genetic accommodation broadly, genetic assimilation specifically).*

Validity

*Based upon my expertise (see below for more information), I found the data interpretations to be valid and compelling. Not only were the results of individual approaches valid, but the fact that multiple, complementary approaches reinforced a variety of conclusions was especially convincing. For example, the potential role of *spr-5* in genetic assimilation was established by ablating the gene, blocking its product, and through artificial selection.*

Significance

For more than a century, and increasingly in the past few decades, a variety of theories have been proposed regarding whether and how non-genetic phenomena such as phenotypic plasticity and trans-generational epigenetic modifications might constrain, promote, or otherwise modify the trajectory of adaptive evolution. Accumulating empirical evidence has supported some of these ideas, which are the subject of a major debate within Evolutionary Biology (i.e., whether the Modern Synthesis should be revised as the Extended Evolutionary Synthesis to account for these less traditional mechanisms). However, few, if any, papers that I can recall address these issues in such a thorough and detailed way.

This manuscript describes a series of complementary studies and experiments through which several of these important threads are woven together with convincing empirical support. These include the roles of genetic assimilation of previously plastic responses, trans-generational inheritance of epigenetic modifications, and the potential for non-genetic variation to act as evolutionary capacitor. In my view, this is an exceptional paper that is likely to become a classic study of the mechanistic basis genetic assimilation, epigenetic inheritance, and their evolutionary consequences. The integration of bioinformatics, comparative transcriptomics, direct genetic manipulation, morphometrics, and experimental evolution paint an extraordinarily detailed and compelling picture of how genetic assimilation has likely repeatedly occurred in this genus of nematodes, and how the mechanism of genetic assimilation might act as a capacitor for further evolutionary change.

Data and Methodology

I judged the methods to be elegant, logical, and mutually supportive. The data appear valid and reliable, insofar as my expertise allows me to fully evaluate them. While I do not directly engage in molecular approaches in my own lab, I collaborate biologists who do, and am confident in my ability to evaluate the results of molecular studies with respect to questions of mechanisms of evolutionary and trans-generational inheritance. I cannot comment on the validity of the molecular methods, but the interpretation of their results seem sound. I do have expertise in morphometric methods, and I do find these approaches and results to be valid.

Analytical approach

The analytical approaches are valid.

Clarity and Context

The text is well-written and the figures and tables are clear, thorough, and should be accessible to a wide audience. This clarity is especially important given that this manuscript uses a variety of integrative approaches, so any given reader might not be expert in every aspect of the paper.

References

The references cited are appropriate in that they support the broad theoretical basis of the manuscript, as well as the methodological approaches.

Reviewer Expertise

My expertise is in the role of plasticity in evolutionary adaptation, particularly with respect to the evolution of morphology. I collaborate on projects involving bioinformatic and genomic approaches, so I feel confident in my ability to evaluate the rationale and results of all aspects of the manuscript. However, I do not myself use molecular techniques, so I am not an expert in the specific methodologies by which these results are generated.

We thank the reviewer for the thoughtful consideration of our manuscript and for the thorough description of its contents and merit.

Suggested Improvements

I have only minor suggested improvements:

Fig.1A—Is this phylogeny reproduced from reference 7 (Fig. 2 therein), or generated as part of the comparative genomic tests for genetic assimilation? I suspect the phylogeny came from reference 7, which would then be used to identify comparisons of interest for the comparative genomic analysis described in the present study. If that's the case, then the figure legend should cite reference 7.

Our response: The tree was indeed reproduced from two studies, specifically references 11 and 23. We thank the reviewer for catching this. We have added the citations to the legend.

296: change to “were scored” instead of “where scored”

Our response: Done.

298: I interpret “under standard laboratory conditions” to indicate that individuals from these three strains were raised in conditions that typically generate low proportions of the Eu morph. Can the authors cite a reference for their “standard laboratory conditions,” or otherwise describe them briefly? Given that the authors are investigating a polyphenism, the details of the environmental conditions are important.

Our response: We have added a citation for these conditions and added a description of them: “...on 6-cm plates with nematode growth medium agar, fed a lawn of *Escherichia coli* OP50 grown in 300 μ l L-broth, and maintained at 20-25 °C.”

306: What R version, and which R package(s) were used?

Our response: We have added, “...as implemented in the R (version 4.0.1) package lme4.”

Reviewer #2 (Remarks to the Author):

The article 'A regulator of nongenetic inheritance mediates the evolution and loss of plasticity' by Levis and Ragsdale reports the finding of a histone tail modifying enzyme (spr-5) that is associated with morph-biased gene expression between two types of feeding morphology and that has been under positive selection in two lineages that are fixed for one type of feeding morphology. Since spr-5 is also a known and demonstrated mediator of epigenetic inheritance, the authors interpret spr-5 as an “epigenetic bridge” that enables (transgenerational) plasticity and that is the target of selection during genetic assimilation.

Our response: We thank the reviewer for thoughtful consideration of our manuscript and for stimulating the additional work that has enriched it.

The manuscript is addressing an interesting idea and presents some suggestive results that are broadly consistent with these ideas. However, the presented data is also consistent with a number of alternative interpretations (see below) that are not sufficiently acknowledged, and conclusion do not rigorously follow from the results. Several additional lines of evidence would be require to makes this a truly convincing piece of work. For example, more information on the function of spr-5 in the lineages that are fixed for one morph would be desirable and strengthen the case. I have several major criticisms about the rationale presented in the study and the soundness of the interpretations.

Our response: We provide new data to address the specific comment here (i.e., function of *spr-5* in lineages fixed for one morph). Specifically, we tested the effects of *spr-5* on mouth morphology in a *P. pacificus* line fixed for a single (Eu) morph. Because CRISPR/Cas9-mediated changes are not yet possible in other species fixed for one morph, we mimicked this effect using an artificially “assimilated” line in a species where we do have such tools. Our results were that, even in the absence of plasticity, *spr-5* has an effect on the mouth morphology, which we have quantified and present in our new Fig. 3. We have also added a corresponding section to the manuscript (“*spr-5* influences mouth morphology...””) to describe these results and their impact. We conclude that, besides affecting plasticity itself, *spr-5* influences the trait apart from the polyphenism network *per se*.

Given this finding, we then performed additional experiments to determine the effects of *spr-5* on plasticity itself, specifically to distinguish whether *spr-5* is important during assimilation, after it, or both. Our study already reported the gene’s influence on the polyphenism threshold (i.e., morph-bias) under standard environmental conditions, but we wanted to know if the *degree* of response to induction cues are affected in mutants. First, we compared the response of *spr-5* mutants to liquid culture, which comprises a set of nutrient conditions known to induce the St morph (Werner *et al.* 2017 *Sci Rep*). We found that the degree of change, or plasticity, was significantly reduced compared to the wild-type (our new Fig. 2A). Second, we performed an experiment to test the effects of *spr-5* when exposed to an ecologically more immediate cue, starvation. Because starvation induces the Eu morph, and the *P. pacificus* reference strain (PS312) is Eu-biased, we constructed a new mutant line by CRISPR/Cas9, specifically to ablate *spr-5* in a *P. pacificus* strain that is normally St-biased in laboratory culture. We then starved these new mutants compared to *spr-5*(+/+) individuals in the same genetic background (our new Fig. 2B). Corroborating our liquid-culture experiment, *spr-5* defects eliminated the induction of the Eu morph in response to starvation. Together, our new data are consistent with the possibility that H3K4 di/monomethylation could result in new morphological variants both during and after the process of genetic assimilation, which we explain in our revision.

In response to the more general comments here, we have clarified additional portions of the text to make phylogenetic relationships, rationale, and other aspects of the manuscript clearer.

Given the evidence presented, the authors make quite far-reaching statements. For example, in lines 51-53: the authors have established two lines with knock-outs in two switch genes that cause the nematodes to constitutively express one of the two morphs. Using differential gene expression between the two lines, they identify genes with a morph-biased gene expression. Having done this for two Pristionchus species and retaining the overlap of these morph-biased gene, they claim that they identified a ‘core set of environmentally sensitive genes ancestral to Pristionchus’. Calling these genes ‘environmentally sensitive’ is quite a stretch since they really are differentially expressed between two mutant lines, and it is far from given that the transcriptomic signature of a CRISPR-induced morph is the same as that of an environmentally-induced morph. As a side, the reader is never told what environmental cue naturally induces the morphs. Why are not naturally-occurring morphs used to identify a set of environmentally sensitive genes? In addition, using two species with unclear phylogenetic positions (after all the reader is told, those could be very closely related, highly derived sister species) is not strong evidence for the set of ‘genes inferred to be ancestral to the entire genus’ (line 53).

Our response: We agree that the term “environmentally sensitive genes” carries unnecessary and possibly misleading connotations in the context of our manuscript. Our reason for using this term was to be consistent with nomenclature introduced in a pivotal review on adaptive plasticity (Via *et al.* 1995 *TREE*). However, whether the morph-biased genes in our study are environmentally induced or simply controlled by a polyphenism switch (although the latter was the basis for the usage coined in the Via *et al.* review) does not

bear on our study design, so we have simply removed the term and now only refer to “polyphenism-biased genes” (line 59, Fig. 1C). We also agree with the implication that our study design deserves more explanation, so we now give our rationale for using genetic manipulations instead of environmental inductions. We consider it a strength of our system and our approach that we can isolate genes whose expression is regulated by a set of polyphenism switch genes, with environment *removed* as a variable, for several reasons: first, polyphenism induction cues are expected to have wide-ranging effects on an organism; second, multiple induction cues can each have idiosyncratic effects on gene expression; third, induction cues are often not (and, in *Pristionchus* species, definitely not) zero-sum in their ability to convert phenotypes, which can only be verified at the phenotypically relevant stage, i.e., past the developmentally relevant window of the polyphenism switch. To summarize this, we have added the following text to the Results:

“Our approach used genetic perturbations in common genetic backgrounds and environmental conditions, which allowed us to distinguish genes associated with the polyphenism *per se* from others whose expression might also be variously influenced by environmental cues [Forsman 2015 *Heredity*]. Further, this approach excluded effects that might be idiosyncratic to individual cues, which in *Pristionchus* include crowding, starvation, and other metabolic influences [Bento et al. 2010 *Nature*; Bose et al. 2012 *Angew Chem*; Werner et al. 2017 *Sci Rep*].”

Additionally, we have added some text to the Methods to supplement this change, specifically to the point on sequencing developmentally relevant stages, citing an article where we more fully explain the logic of the approach (Bui and Ragsdale 2019 *Mol Biol Evol*).

Regarding the phylogenetic positions of *Pristionchus* species: although our tree (Fig. 1B) is formally correct in identifying the most recent common ancestor of *P. pacificus* and *P. fissidentatus* as the most recent common ancestor of all *Pristionchus* species, we realize that this tree could be made more informative with respect to relationships within *Pristionchus*. Therefore, we have expanded the tree figure to show major clades of *Pristionchus*, which we had omitted for the sake of brevity.

-selection-based analysis (lines 54-70): This section would greatly benefit from more details and it's very hard to interpret how convincing the data actually is. How many genes were identified as positively selected in the two species that allegedly recently lost plasticity? How many genes in the two species that lost plasticity longer ago? Also, no evidence is provided that the two events of plasticity loss are actually on different time scales since the phylogeny in Fig. 1 is not dated.

Related to this, I am not convinced that looking for genes that have experienced positive selection in the past is the right type of analysis here. Why would a gene with a morph-biased expression be under positive selection in a lineage constitutively expressing one of the morphs? Given the direction of morph-biased expression, a really important gene might as well have become pseudogenized or lost its function in the lineage that fixed the morph in which it's not expressed. At the very least, the analysis should be performed directional, meaning to look for overlap of type-A-biased genes that are under positive selection in species only exhibiting type-A (but not type-B) morphologies. At present, this analysis seems poorly thought-through, superficially conducted and insufficiently described.

Our response: We agree that this part of the study needed greater clarification, and we appreciate that this was pointed out. We have made several changes to address this issue.

First, as the reviewer notes, we do not know if the two assimilation events differ in geological time. Our assumption of relative ages was based on operational time as inferred previously (Susoy et al. 2015 *eLife*), but we concede that we do not definitively know the dates of these independent events. Therefore,

we have changed the wording to indicate that the assimilation events are simply independent, which is the feature relevant to our study design.

Second, we have expanded the presentation of genes for each species that had evidence of positive selection (Extended Data Table 1).

Third, we have added a description of the morphologies of the assimilated species: only *P. bucculentus* is clearly fixed for the Eu morph, while the other three species possess morphologies that are novel with respect to other diplogastrids and therefore of ambiguous homology (lines 66–71, 79–83). This rapid evolutionary change in form following assimilation was predicted based on theory presented by West-Eberhard (2003) and references therein. That this was demonstrated statistically in Diplogastridae (Susoy et al. 2015 *eLife*) is, in our opinion, a strength of the system, as it implies rapid evolution of genes mediating alternative forms. Given the morphological divergence in lineages that have assimilated a single morph, we opted to take an unbiased (rather than directional) approach toward identifying genes experiencing positive selection. This approach sought to identify genes associated with assimilation *per se*, rather than a particular morphology. That said, we did look at the morph-bias of these genes and did not observe overrepresentation of one morph or the other being subject to positive selection.

Fourth, we have added new data. We agree that we might expect to find relaxed selection on those genes associated with the non-fixed morph, which is predicted by theory (West-Eberhard 2003; Van Dyken and Wade 2010). More generally, plasticity can also be lost because of relaxed selection or drift (Pál and Miklós 1999; Pigliucci et al. 2006; Masel et al. 2007; Vigne et al. 2021). Therefore, to further address these points, we have now included an assessment of relaxed selection (using the RELAX tool in HyPhy) to infer the intensity of selection on a foreground set of genes relative to a reference set. The results of this test are not mutually exclusive with those of positive selection using aBSREL: the former test compares relative strengths of selection among sets of genes, while the latter determines if including a parameter for positive selection for a proportion of sites along a branch improves the model fit. Using assimilated species as the foreground and the rest of each tree as the background, we did not find a pervasive signature of relaxed selection for a particular class of morph-associated genes (Eu or St), and we report these data in our revision. However, we did observe that a restricted set of the polyphenism-biased genes (five total) experiencing positive selection *also* exhibited relatively relaxed selection compared to the rest of their tree. Excitingly, and strengthening our conclusions, *spr-5* was in this set of five genes. Convergence of both tests on this same gene suggests that accumulated variation in the gene, whether owed to positive selection, relaxed selection, or both, is an important contributor to genetic assimilation. Indeed, our functional tests confirm this gene's importance. The revised text reads:

“We next tested if the strength of selection acting on genes in each of our assimilated species has been relaxed relative to the strength acting on polyphenic species. Our rationale was threefold: first, plasticity can be lost due to selection or drift [Pál and Miklós 1999; Pigliucci et al. 2006; Masel et al. 2007; Vigne et al. 2021]; second, genes not associated with the phenotype that was canalised during genetic assimilation might have experienced relaxed selection following plasticity's loss [West-Eberhard 2003; Van Dyken and Wade 2010]; third, this analysis would identify additional targets whose evolutionary history is similar during independent cases of assimilation. As with our inference of positive selection, we observed a limited set of once polyphenism-biased genes (18 total) with signatures of relaxed selection across both assimilation events (Extended Data Table 3). Moreover, five of these genes were also present in our set of shared genes experiencing positive selection. This suggests that accumulated variation in these genes—whether by selection, drift, or both—is an important contributor to genetic assimilation and its associated diversification. These comparisons thus point to a set of once morph-biased genes that were potentially key targets of evolution associated with multiple, independent losses of developmental plasticity in Diplogastridae.

Among the five targets reporting signatures of selection in both analyses was the histone H3-di/monomethyl-lysine-4 demethylase gene *spr-5/LSD1/KDM1A...*”

In summary, we thank the reviewer for stimulating us to further dig into our results and thereby enrich the manuscript's thesis.

-the authors selected spr-5 among the 19 genes that were identified as having a morph-biased gene expression in Pristionchus and having experienced positive selection in at least one species of the two lineages that lost plasticity. What about the other 18 genes? Given their equally strong association with the morphs as spr-5, it is well conceivable that their ablation might also have caused differences in the morph-expression, similar to spr-5. What would this mean for the interpretation of the role of spr-5?

Our response: We agree that the other genes showing selection signatures might have effects on the phenotype. This is consistent with theory, which predicts that the process of genetic assimilation (and genetic accommodation more generally) entails the accumulation of plasticity modifiers that stabilize and refine the adaptive phenotype. Therefore, multiple loci would be expected, and using these other genes to further test predictions of assimilation theory is an active area of our lab's research. However, we do not see how phenotypic effects of one or more of those other genes should influence our interpretation of SPR-5 function, unless one of those other genes was a direct, physical interactant with SPR-5 (and none are, based on their annotations). We chose to pursue *spr-5*, from the 20 we had found, based on a list of other criteria that we describe in the text. However, this choice is further justified now that we have narrowed our list of candidates by adding analysis of relaxed selection (please see above). There are additional genes that still fall into this stringent list, and we have added text (lines 85–88) describing that these changes, beyond *spr-5* should be expected according to assimilation theory.

-arguably using a loss of spr-5 function to test whether this modification could allow directional selection upon new morphologies as observed in species with fixed morphs is not ideal. Ablating gene function is rather the opposite of mimicking the effect of positively selected alleles, and therefore I am not sure what the biological relevance of a spr-5-negative organism in the context of this study is. Since the logic here is reverse to what I would have thought is relevant to test, I do not follow the interpretation of the authors, and the far-reaching conclusion that they draw based on this experiment.

Our response: We thank the reviewer for raising this point. However, we disagree that loss of function necessarily means maladaptive. For example, adaptive evolution of morphology in mice (Hoekstra et al. 2006 *Science*) and sticklebacks (Chan et al. 2006 *Science*) both involved loss-of-function mutations. Similarly, positive selection can also be detected with adaptive loss of function, specifically in enzymes (Bailly et al. 2003 *PNAS*). Thus, ablation can be an extreme way of assessing modified gene function related to adaptive evolution. We now point this out, and reference the precedent for it, where we describe our rationale for this experiment. Furthermore, theory predicts that adaptive peak shifts may occur via genetic drift of genes controlling epigenetic variability (Pál and Miklós 1999). Given that *spr-5* is such a gene, and that we have evidence for both positive selection and relaxation of selection intensity on this gene, we think that genetic ablation is a useful proxy for assessing functional modifications this gene might have experienced during evolution.

-minor question: are the 838 genes identified as differentially expressed in both species concordantly differentially expressed? I.e., are the 838 genes up- and down-regulated in the same morphs between the two species? If not, those genes should be filtered to retain only those with consistent expression changes since the discordant ones are probably false-positives.

Our response: Yes, we only retained those genes that were concordant in morph bias. The legend of Figure 1 mentioned this, but we have now also made it clear in the methods.

Overall, I find the presented ideas interesting and the presented results suggestive, but there are several important gaps in the logic that unfortunately mean that the conclusions are not fully supported by the presented evidence.

Our response: We believe that our revisions, especially the suite of new experiments and analyses we provide, have addressed this concern.

REVIEWER COMMENTS

Reviewer #2 (Remarks to the Author):

Levis & Ragsdale have revised their manuscript entitled 'A regulator of nongenetic inheritance mediates the evolution and loss of plasticity' in the light of my and others' comments. The authors have made some modifications to the MS and added additional analyses in an attempt to strengthen their arguments. However, I find that substantial weaknesses remain, and several claims are not sufficiently supported by the results.

The most problematic claim is that *spr-5* has played a role in assimilation (and exaggeration) or feeding morphologies in nature (abstract, lines 7-8). In response to my previous comment, the author have added a test for relaxed selection in addition to the test for positive selection. Unfortunately, addition of this test exposed severe problems with these analyses: the gene *spr-5* is identified as both under diversifying/positive as well as under relaxed selection in the very same lineages (Pbu + Pel + Ltex). This result is simply not possible and probably a sign of some deeper issue with these analyses. It is also more than unfortunate that the authors have chosen to show these results only in the Extended Data, since the results of these analyses would be highly relevant for the reader.

In addition to this major issue, the set-up of the selection analyses are convoluted and the reporting is not logical: rather than analyzing all data (lineages) at once, the authors chose to first use a taxonomically-restricted dataset, and then a larger dataset. However, the results of these analyses are not sufficiently reported, and the rationale for this 2-step-approach is not justified. It is also unclear if the morph-biased genes identified in these selection analyses are more than one would expect to find based on chance alone (Extended Data Table 1 should contain this information).

Since I do not think that the presented results supports the main claim of the paper, namely that evolutionary modification of *spr-5* has 'mediated the evolution and loss of plasticity' (title), I recommend the authors to reconsider the framing of the study. Showing that *spr-5* has an effect on the expression and environmental sensitivity of mouth morphologies in lab strains of 2 species is not sufficient to conclude that this gene has actually played a role during the evolutionary process. Large parts of the manuscript (abstract lines 14-18) are mere speculations and reach far beyond the actual results reported in the study.

In order to strengthen the role of *spr-5* in mediating evolution and loss of plasticity via nongenetic inheritance, the following evidence would be required:

-the role of *spr-5* in evolutionary lineages with modified feeding morphologies should be investigated in more detail. Part of this could be rigorous selection test, but also analyses of *spr-5* expression levels. The current paper hones in on coding-changes as the only possible mechanism of change in a gene's function over time, but expression levels are equally plausible mechanisms (perhaps more so for plasticity-led evolution).

-the suggested mechanism of transgenerational inheritance via di-methylation of H3K4 marks should be substantiated. The fact that Bizine induces a phenocopy of the *spr-5* mutant is suggestive, but actual analyses of histone tail modifications (e.g. through CHIP-seq) would be required to robustly establish inheritance of histone modification as the mechanism of transgenerational persistence of the phenotype.

-to make the argument that *spr-5*-mediated epimutations fuel transgenerational shifts in plasticity and morphology (last section of results), the selection experiments should be repeated with the appropriate control, namely an inbred wild-type strain. The authors refer to Ref. 15 saying that selection on inbred lines fails to drive any shift, but this test should be repeated as replicate along with the *spr-5* lines and shown alongside in Fig. 4B.

Minor comment:

-line 224: 'for that phenotype' should be delete. The fact that *spr-5* plays a role in DNA damage repair might facilitate accumulation of new variation in general, but not specifically for that phenotype.

RESPONSE TO REVIEWERS' COMMENTS

Reviewer #2 (Remarks to the Author):

Levis & Ragsdale have revised their manuscript entitled 'A regulator of nongenetic inheritance mediates the evolution and loss of plasticity' in the light of my and others' comments. The authors have made some modifications to the MS and added additional analyses in an attempt to strengthen their arguments. However, I find that substantial weaknesses remain, and several claims are not sufficiently supported by the results.

*The most problematic claim is that *spr-5* has played a role in assimilation (and exaggeration) or feeding morphologies in nature (abstract, lines 7-8). In response to my previous comment, the author have added a test for relaxed selection in addition to the test for positive selection. Unfortunately, addition of this test exposed severe problems with these analyses: the gene *spr-5* is identified as both under diversifying/positive as well as under relaxed selection in the very same lineages (Pbu + Pel + Ltex). This result is simply not possible and probably a sign of some deeper issue with these analyses. It is also more than unfortunate that the authors have chosen to show these results only in the Extended Data, since the results of these analyses would be highly relevant for the reader.*

Our response: We appreciate that the reviewer has raised this issue, which with insufficient explanation may seem paradoxical at face value. Therefore, we have added further detail and precision to the Methods as well as the Results, where we now explain the logic of our approach and the benefits we expected it to bring. In short, the two selection-detecting methods we used (aBSREL, RELAX) do not operate the same way, allow different sites across a gene to drive a signal, and are both a way of detecting significant diversification. As such, it is not uncommon for the results of these two methods to overlap, including where they have identified rapidly evolving, adaptive genes (e.g., Schneider et al. 2019 *BMC Genomics* 20:1010; Berger et al. 2021 *BMC Ecol Evol* 21:48; Pirri et al. 2022 *Heredity* 129:317; Bilyk et al. *Genome Biol Evol* 15:evad049). To clarify our logic, as we now fully flesh out in both the Methods (lines 403–415) and the Results (95–99), we explain it here.

Specifically, the aBSREL method for detecting positive (diversifying) selection focuses on determining some proportion of sites along branches of interest have evidence of these forms of selection. It does so by comparing models that do or do not allow multiple ω (dN/dS) rate classes and where ω is (dis)allowed to be greater than 1. Thus, this test determines if some proportion of sites bear a signature of diversification. In contrast to the operation of aBSREL, RELAX evaluates the relative intensity of selection acting on a set of focal branches compared to other branches. This intensity is indicative of selection “pushing” ω values toward 1 in the focal branch(es) compared to the reference branches. So, even if a branch has evidence for positive selection, the intensity of that signal might be lower than other branches, especially when looking across the whole gene. Therefore, both methods can detect a rise in ω , and thus potential signals of diversification, but in different ways. In the case of *spr-5*, there was evidence of episodic, diversifying selection within the focal branches and a relatively weaker intensity of selection compared to the rest of the tree. Both of these signals point to increased diversity in this gene in the assimilated lineages, as predicted by theory.

Beyond our text additions, we also provide new data on this topic. Specifically, we performed an *ad hoc* analysis, using a different program (codeML), to detect individual sites under selection. This type

of analysis can identify sites with relatively strong evidence of positive selection. Having identified these sites, and other sites where amino-acid identity diverged from the common ancestor of *P. bucculentus* and *P. elegans*, we assessed, using a protein modeling tool, the potential changes such variants would have on the encoded protein. Our results showed that variant sites showing (or not showing) positive selection were not biased toward particular motifs or domains of SPR-5, indicating the potential of both to potentially drive functional changes in the protein. The results are reported on lines 114–125 and in Table 3.

In summary, we appreciate the reviewer's comment, which has encouraged us to explain, justify, and explore our results. Consequently, these efforts have resulted in a more intuitive presentation of our findings.

Regarding the two data tables, we simply placed them in the Extended Data for the sake of efficiency. Pending the discretion of the editor on this issue, we have moved those tables to the main document.

2) In addition to this major issue, the set-up of the selection analyses are convoluted and the reporting is not logical: rather than analyzing all data (lineages) at once, the authors chose to first use a taxonomically-restricted dataset, and then a larger dataset. However, the results of these analyses are not sufficiently reported, and the rationale for this 2-step-approach is not justified. It is also unclear if the morph-biased genes identified in these selection analyses are more than one would expect to find based on chance alone (Extended Data Table 1 should contain this information).

Our response: We thank the reviewer for highlighting this oversight in our explanation. The reason we performed two analyses instead of one was simply to maximize the number of orthogroups we could identify for each "assimilated" lineage. We now explain this in the Methods (lines 421–423). As a result, the taxonomically restricted dataset included more genes, giving us more power (although, obviously, *spr-5* would have still been included in the analysis if we did one rather than two analyses).

Regarding the number of morph-biased genes bearing such signals compared to genome-wide: in general, there were not more than expected by chance when looking at a particular morph or at being polyphenism-associated. Although these descriptive statistics do not bear on our study design, we now include this information in the table (now Extended Data Table 1), as suggested.

*Since I do not think that the presented results supports the main claim of the paper, namely that evolutionary modification of *spr-5* has 'mediated the evolution and loss of plasticity' (title), I recommend the authors to reconsider the framing of the study. Showing that *spr-5* has an effect on the expression and environmental sensitivity of mouth morphologies in lab strains of 2 species is not sufficient to conclude that this gene has actually played a role during the evolutionary process. Large parts of the manuscript (abstract lines 14-18) are mere speculations and reach far beyond the actual results reported in the study.*

Our response: We have seriously considered this recommendation, and we agree that the framing of the paper could be shifted to more directly represent our results. Consequently, we have changed the title and relevant parts of the Abstract (now lines 7–8, 10–13) and Introduction (27–31). Moreover, we have greatly expanded the Discussion. Because our Discussion was relatively brief before, we concede that our emphasis of this speculation may have been out of balance when considering the sum of our

findings. Also, we agree that our results do not provide direct evidence for what happened in evolutionary history, but rather that history was used as an unbiased screen for genetic factors mediating plasticity evolution and loss. Therefore, we more clearly identify that part of the Discussion as speculation, which we retain because it places our results within the wider body of theoretical work on these issues. Besides this change, we have also added three paragraphs to the Discussion that highlight other major impacts of our results (as well as another paragraph before our Conclusions). The first is that they introduce a specific mechanism of transgenerational, nongenetic inheritance to the study of plasticity, fulfilling an outstanding but conspicuously unanswered prediction of the field. Second, we describe the impact our findings have on how specific genes can influence both plasticity and ultimate trait values another long-standing controversy in the field. In summary, we are grateful for this comment, which has encouraged us to more thoroughly highlight the implications of our study.

In order to strengthen the role of spr-5 in mediating evolution and loss of plasticity via nongenetic inheritance, the following evidence would be required:

-the role of spr-5 in evolutionary lineages with modified feeding morphologies should be investigated in more detail. Part of this could be rigorous selection test, but also analyses of spr-5 expression levels. The current paper hones in on coding-changes as the only possible mechanism of change in a gene's function over time, but expression levels are equally plausible mechanisms (perhaps more so for plasticity-led evolution).

Our response: This comment invokes false dilemma, as we do not make the claim that coding changes are the *only* possible mechanism of change. As with any study, we addressed one facet of a multifaceted phenomenon, and that focus does not rule out the alternatives. We agree that there may be other, equally plausible mechanisms, but these are by no means mutually exclusive with our findings and would instead probably complement them. In fact, *spr-5* itself mediates gene-expression through its activity as a chromatin modeler. To highlight this in our revision, we have added a new paragraph to the Discussion that acknowledges (lines 257–264) the potential importance of gene-expression changes in plasticity evolution, specifically those that *spr-5* manipulation will allow one to detect.

-the suggested mechanism of transgenerational inheritance via di-methylation of H3K4 marks should be substantiated. The fact that Bizine induces a phenocopy of the spr-5 mutant is suggestive, but actual analyses of histone tail modifications (e.g. through CHIP-seq) would be required to robustly establish inheritance of histone modification as the mechanism of transgenerational persistence of the phenotype.

Our response: We agree that ChIP-seq analysis will ultimately improve our understanding of the mechanism of *spr-5* in the transgenerational memory in the polyphenism. The improvement would come from establishing if and where the H3K4me1/2 changes occur. Therefore, our revision acknowledges the promise of such an approach in our revised Discussion. In fact, this is a major direction of our lab's research, and to do the needed experiments with the thoroughness they warrant would be far out of the scope of the present study. That said, we do not agree that ChIP-seq evidence is required for the claims that we make. Throughout the manuscript, we explain that the methyl marks are the likely mechanism, and we have combed through our revision to make sure our wording on this is consistent with this claim (e.g., on lines 184, 296–297). Even this caution, in our opinion, is quite conservative given that the H3K4 mono- and dimethylation activity of *spr-5*, of which *P. pacificus* has a

single ortholog, is highly conserved in all other animals where it has been studied. This evidence is in addition to the Bizine assay and our ability to drive phenotypic divergence in genetically inbred lines, which together establish robustly that nongenetic information is being transmitted across generations and, as the reviewer notes, is suggestive that this is via H3K4me2. In any case, whatever the mechanism of *spr-5* loss in our transgenerational experiments, they clearly show the gene's effect. In summary, while we agree that ChIP-seq will be an interesting follow-up to this study, we argue that it is not necessary to support our thesis here.

-to make the argument that spr-5-mediated epimutations fuel transgenerational shifts in plasticity and morphology (last section of results), the selection experiments should be repeated with the appropriate control, namely an inbred wild-type strain. The authors refer to Ref. 15 saying that selection on inbred lines fails to drive any shift, but this test should be repeated as replicate along with the spr-5 lines and shown alongside in Fig. 4B.

Our response: This suggestion asks us to repeat the established finding that *P. pacificus* laboratory strains lack selectable variation. In response, we kindly point out that the inability to select on plastic phenotypes, as demonstrated by Bento et al. (2010; i.e., the reference referred to here), was already performed with appropriate replication. Besides this, it is well established that strains of *P. pacificus* are highly homozygous, and we now supply additional references in that section of the Results (lines 200–201) so that readers are fully aware of this. In case it is assumed that *de novo* mutations might be selected, we also point out that this species has a mutation rate typical of many animals (on the order of 10^{-9}), making it highly unlikely (in practice, impossible) that any phenotype can detectably selected from new mutations over 10 generations. Therefore, we believe that the experiment suggested here would be an unnecessary formality and not add to the study. For this reason, we have declined to add this experiment to our revision.

Minor comment:

-line 224: 'for that phenotype' should be delete. The fact that spr-5 plays a role in DNA damage repair might facilitate accumulation of new variation in general, but not specifically for that phenotype.

Our response: The change has been made.

REVIEWER COMMENTS

Reviewer #3 (Remarks to the Author):

This paper has apparently been through several rounds of revisions. The authors have responded to the last round of comments by adding analyses, modifying and adding text, including a substantially expanded Discussion. There is a lot of good things to say about this work, and the authors have responded appropriately to most of the comments. Nevertheless, I must admit that I share the scepticism of reviewer 2 regarding the robustness of some key results, and I also understand the reviewer's apparent frustration over the presentation and interpretation of the results.

My take on this is that the authors try too hard to fit their work into a framework for the relationship between plasticity and evolution that the reported study shed little light on in practice. As a result, there is a mismatch between conceptual setting and study design, and between results and interpretation. This is perhaps also why the writing seems convoluted and full of unnecessary jargon (some of it used in an awkward way). I have listed some specific examples below to illustrate this point. The authors will naturally be frustrated at some of these comments, but hopefully they can still be helpful (although I of course acknowledge that the style of presentation is a very subjective matter).

On the whole, this is a fascinating set of results. Unfortunately, the inappropriate focus and excessive jargon will likely frustrate readers who still find the relationship between plasticity and evolution confusing or controversial. As a result, the current version of the paper could at worst be dismissed or ignored, and serious readers may fail to grasp the real importance of the work.

More detailed comments are as follows:

Lines 3-5: "Where such change precedes genetic change, the exploration of alternative phenotypes may promote a trait's evolution, especially when those phenotypes' appearance is followed by plasticity's loss, or genetic assimilation."

Lines 20-21: "Evolutionary change in a trait's developmental sensitivity to the environment has long been predicted to influence the evolution of the trait itself"

Comment: As pointed out above, the broad view that plasticity can promote evolutionary adaptation and diversification does not feel appropriate for the present study.

What this study is attempting to do is to (i) identify genes associated with loss of polyphenism using a combination of transcriptomic and comparative genomic analyses; (ii) verify the functional consequences of this gene by knocking it out; and, motivated by the identity of the gene (iii) test if it is possible to get a response to selection on morphs in a loss-of-function strain.

These are excellent aims, but why try to fit this into 'plasticity-led evolution' scenarios? The polyphenism/polymorphism is ancestral and the morphology apparently both refined and adaptive. In contrast, the setting the authors refer to in their Abstract and Introduction brings to mind scenarios of evolution via environmentally induced phenotypes as discussed by West-Eberhard and others. Of course, there is a sense in which loss of polyphenism is 'genetic assimilation', but what good can come out of using the term in this context (i.e., loss of one adaptive morph rather than fixation of a phenotype that appeared – without first being selected – in an extreme environment)?

I recommend that the authors consider that their choice of presentation and context could cause many readers to be confused, and that another group of readers will be annoyed when a study like this – that does not address a controversial topic in itself – is put (mistakenly in my view) in a context that is contentious, without any obvious pay-off.

Lines 16-18: "Our findings thus point to the plausibility that modified spr-5 function sources morphological changes ahead of genetic change, giving mechanistic insight into how traits are modified as they traverse the continuum of greater to lesser environmental sensitivity."

Comment. An example of complex/convoluted wording that borders on the misleading (I refer to the "sources morphological changes ahead of genetic change" part: the work documents the effects of spr-5 mutants, and is not a test of a plasticity-first scenario or similar).

Lines 24-27: "Beyond the fixation of allelic variants expected to occur during assimilation, and which have been identified in some cases^{5,6}, theory suggests that epigenetic mechanisms and nongenetic inheritance, because they might offer a bridge to genetically inherited change, should be key factors in facilitating genetic assimilation⁷⁻¹⁰."

Comment: This is a bit awkward here in the Introduction, because the study was not originally set up to test for the role of epigenetic mechanisms and non-genetic inheritance. This line of investigation follows from the identification of spr-5 as a putative candidate (although there is arguably not much 'epigenetic' in this paper, as reviewer 2 notes). Arguably it is a matter of preference, but given the overall feeling that there is a mismatch between how the paper is presented and what was done, this wording does come across as a bit too much ad hoc reasoning.

Line 40. Spell out what the "signature of evolution associated with plasticity's loss" should be. Presumably relaxed selection?

Line 45-46. The use of genetic mutants to identify genes that are differentially expressed between morphs is creative, but it is also quite unexpected to not include wild-type morphs. The candidate genes identified here dictate all the downstream work, so it is peculiar to not demonstrate that these signatures are consistent with wild type phenotypes. Of course there is the risk of environmental effects 'blurring' the differences, but the two 'natural' morphs could still have been included since a given environment does not induce 100% of either morph. Note that, if one considers the lack of contrast here odd, the identification of genes under selection will come across as less-than-solid.

Lines 57-59. The authors will obviously be aware that transcriptomes of whole worms are associated with all kinds of problems when it comes to interpreting the results in terms of mouth morphology. I assume it was necessary to do this to get enough tissue. Yet, it would be good to keep in mind that any conclusions from this work needs to make the assumption that the relevant gene expression differences were identified. Some words of caution could be appropriate here or elsewhere (e.g., Discussion).

Lines 66-70. Here it would be useful to say what morph P bucculentus has, this helps to interpret the results later.

Line 72. Transcriptomes from the 41 species mentioned above, I assume?

Line 74: "...thereby defining evolutionary targets associated with the assimilation of once-plastic morphology"

Comment: Very awkward wording, why not just say loss of polyphenism?

Lines 76-86: This whole section of results on the Diplogastridae are a bit difficult to make sense of and it would be helpful to include some details that help the reader see the difference between the two clades (transcriptomic vs genomic data; note that the latter is not actually 'genome-wide', or else it could not be compared to the former) and how the results are combined.

Lines 86-89: "Consistent with the expectation that genetic assimilation, and the evolution of plasticity more generally, involves the accumulation of variants that modify, stabilize, and refine an adaptive plastic response^{2,24,25}, we found multiple loci to contribute to assimilation's evolutionary signature."

Comment: This is a very dense and convoluted way of saying that you identified several candidate genes.

If I understand the authors logic, they did not strictly speaking identify any candidates at all, since the numbers are not greater than expected by chance (see comment and response to reviewer 2).

The text does not make clear if the overlap between comparisons (i.e., clades and gene expression vs sequence data) are greater than expected by chance. Even if not, perhaps one can say that those that do overlap may show something interesting and therefore are candidates worth exploring (after all, we can never know from the results of these analyses which candidates that are false positives, and the authors pursue a single gene based on its known function, not the strength of association). I believe some readers will take this issue quite seriously so it is not a good strategy to sweep it under the carpet.

Lines 101-104: As pointed out in the SI (following a response to reviewer 2), "relaxed purifying selection and episodic diversifying selection are both consistent with positive shifts in ω ." This is true, but sadly it also means these numbers should be interpreted very carefully. Maybe in practice all that may have happened is just relaxed selection, for example? Overall, since it is unclear if the candidates derived from cross-validation (between clades and between gene expression and sequence data) are statistically supported candidates, one does wonder how much weight we should put on *str-5* at this point...

Lines 115-118: Why use ancestral reconstruction – which must be tricky given fast episodic and relaxed selection – rather than compare to both species directly?

Lines 123-125: Why is this interpreted as 'complex evolutionary history'? Also, the evidence is consistent with relaxed selection, but is it also possible that you just detect the signatures of a large target of neutral sequence change?

Lines 136-138: "...with and without a *spr-5* loss-of-function allele, induction of the Eu morph, which was observed in *spr-5* wild-type individuals (estimate = 2.78, Z = 7.19, P = 1.34 x 10⁻¹²), was abolished in mutants (estimate = - 0.48, Z = 1.91, P = 0.11; Fig. 2B)."

Comment: This reporting stands out because the appropriate test is the morph ratio in the wild-type vs mutants, not separate tests for the two groups. The methods give the impression that the statistics were performed by comparing frequencies between wild-type and mutant, but it is not entirely clear how this was done, what was the level of replication, and so on. I encourage the authors to make sure it is transparent in the main text that the right groups have been compared (with other factors included in the analyses, if appropriate).

Lines 145-148: "This uncovering of cryptic or new variation, especially in previously unoccupied regions of morphospace could, in principle, act as an evolutionary capacitor and fuel natural selection. In sum, mutations in *spr-5* affected the regulation and production of a morphology that has rapidly diversified in correlation with its genetic assimilation"

Comment. Again, the wording makes this sound much more complicated than it is. What this demonstrates is that knocking out the gene affects both the incidence of discrete morphs, and the morphology. Maybe this is unexpected (it does not seem like it would be?), but to say that 'knocking out *spr-5* causes phenotypic variation' is arguably easier to understand than the authors' wording. Moreover, given what you are actually doing in the lab, it seems ambitious to extrapolate to consequences over evolutionary time...

Lines 153-155: "We reasoned that if *spr-5* could affect once-plastic morphology even after the completion of genetic assimilation, *spr-5* mutants should have an effect in *P. pacificus* individuals with morph-constitutive development."

Comment: Why is this introducing a different species with a single morph? Why not use the ones that were the focal species in the first set of analyses?

Lines 170-174: "We found that (i) heterozygous F1 showed a significant reduction in the Eu morph compared to a wild-type self-cross, (ii) this epigenetic effect was erased in the F2, which returned to the wild-type phenotype, and (iii) homozygous *spr-5* F3 clones were significantly less Eu-biased than homozygous wild-type F3, showing conversion to the mutant phenotype in exactly one generation."

Comment: This is really quite complex (also, why is it an 'epigenetic effect' and not just a 'maternal effect?'), and the summary is not accompanied by supporting statistics. Consider spelling things out in more detail to help the reader how and why the results lead to particular conclusions.

Lines 165-170: What species is this now?

Lines 189-191: "Finally, we tested whether loss of spr-5 function, and thereby the production of epimutations, would allow directional selection on the plasticity (i.e., morph induction) as well as new variants of the morphology itself."

Comment: This statement contains many assumptions and connections that seem unnecessary and confusing. For one, as far as I can tell, you have not demonstrated that loss of spr-5 function produces epimutations (maybe somewhere else or by others in *C. elegans*?). The second part of the sentence suffers from the general issues of convoluted wording that make everything sound much more complex than it is.

Lines 191-194: "Since assimilated species, which show increased rates of form evolution, also show signatures of increased diversification in spr-5, we hypothesized that modified spr-5 function could allow directional selection upon new morphologies."

Comment: This is really very difficult to follow.

Lines 195-203: Sadly, this selection experiment seems to lack an appropriate control, something that reviewer 2 also picked up on. The authors dismiss the criticism by referring to previous work showing a failure to get a response to selection on these lines, but I am afraid that this misses the point. It is the risk of standing genetic variation (or new sequence variants, perhaps induced by the gene editing itself) in the present strain/population that is an issue. While we know that epigenetic variation exists, can be heritable, and enable responses to selection, it is still the case that demonstrating convincingly that epigenetic variation is the substrate for responses to selection will require appropriate controls (or sequencing of selected lines). As much as I would like to accept the assumption that all of this is epigenetic, in my opinion the authors would be better off were they to avoid such statements and simply refer to selection on the lines. That they do respond COULD mean that spr-5 mutants accumulate heritable epigenetic variation, but demonstrating this will require further work (note that this does not undermine the value of the work as a whole, just again show a mismatch between results and inference in this paper).

Lines 202-207: Again, this reporting seems to imply that the authors did not conduct the appropriate statistical test of differences between groups. I hope they did.

Discussion: Most of the additions are helpful, but please do consider the general comment about over-interpretation –extrapolation of results and context. It would be a shame if readers dismissed this work because of how the authors have chosen to present it.

RESPONSE TO REVIEWERS' COMMENTS

Reviewer #3 (Remarks to the Author):

This paper has apparently been through several rounds of revisions. The authors have responded to the last round of comments by adding analyses, modifying and adding text, including a substantially expanded Discussion. There is a lot of good things to say about this work, and the authors have responded appropriately to most of the comments. Nevertheless, I must admit that I share the scepticism of reviewer 2 regarding the robustness of some key results, and I also understand the reviewer's apparent frustration over the presentation and interpretation of the results.

My take on this is that the authors try too hard to fit their work into a framework for the relationship between plasticity and evolution that the reported study shed little light on in practice. As a result, there is a mismatch between conceptual setting and study design, and between results and interpretation. This is perhaps also why the writing seems convoluted and full of unnecessary jargon (some of it used in an awkward way). I have listed some specific examples below to illustrate this point. The authors will naturally be frustrated at some of these comments, but hopefully they can still be helpful (although I of course acknowledge that the style of presentation is a very subjective matter).

On the whole, this is a fascinating set of results. Unfortunately, the inappropriate focus and excessive jargon will likely frustrate readers who still find the relationship between plasticity and evolution confusing or controversial. As a result, the current version of the paper could at worst be dismissed or ignored, and serious readers may fail to grasp the real importance of the work.

Our response: We thank the reviewer for their thoughtful, constructive, and specific comments on our manuscript. We have taken the reviewer's views to heart and have revised the manuscript, especially its general presentation, along the lines described. Specifically, we have reduced jargon and reframed the manuscript away from an explanation of genetic assimilation and plasticity-led evolution and have instead leaned into how our findings inform mechanisms of plasticity.

More detailed comments are as follows:

Lines 3-5: "Where such change precedes genetic change, the exploration of alternative phenotypes may promote a trait's evolution, especially when those phenotypes' appearance is followed by plasticity's loss, or genetic assimilation."

Lines 20-21: "Evolutionary change in a trait's developmental sensitivity to the environment has long been predicted to influence the evolution of the trait itself"

Comment: As pointed out above, the broad view that plasticity can promote evolutionary adaptation and diversification does not feel appropriate for the present study.

What this study is attempting to do is to (i) identify genes associated with loss of polyphenism using a combination of transcriptomic and comparative genomic analyses; (ii) verify the functional consequences of this gene by knocking it out; and, motivated by the identity of the gene (iii) test if it is possible to get a response to selection on morphs in a loss-of-function strain.

These are excellent aims, but why try to fit this into 'plasticity-led evolution' scenarios? The polyphenism/polymorphism is ancestral and the morphology apparently both refined and adaptive. In

contrast, the setting the authors refer to in their Abstract and Introduction brings to mind scenarios of evolution via environmentally induced phenotypes as discussed by West-Eberhard and others. Of course, there is a sense in which loss of polyphenism is 'genetic assimilation', but what good can come out of using the term in this context (i.e., loss of one adaptive morph rather than fixation of a phenotype that appeared – without first being selected – in an extreme environment)?

I recommend that the authors consider that their choice of presentation and context could cause many readers to be confused, and that another group of readers will be annoyed when a study like this – that does not address a controversial topic in itself – is put (mistakenly in my view) in a context that is contentious, without any obvious pay-off.

Our response: Reflecting upon this general comment, we agree with the reviewer's outline of what our study accomplishes, and we are grateful for the appreciation of our findings' impacts. Our intention was, as the reviewer suggests, to use our results to speculate on the processes of plasticity-led evolution, but we concede that we should more directly contextualize our key findings. Consequently, we have reduced the broad-strokes language about plasticity's role in evolution and adaptation. Our presentation and context, in terms of macroevolutionary loss of plasticity to increased rates of diversification, is now limited to how the work is directly relevant to previous work in this system, particularly since our functional tests make similar connections between plasticity and morphological variation.

Following the reviewer's specific suggestion on terminology, we have changed the term "genetic assimilation" to "loss of plasticity" in many places throughout the manuscript, including in the title and abstract, especially where we think it is important to avoid the connotation with scenarios as the reviewer describes. However, we still include the term "genetic assimilation," which we now define in the first sentence of the main text, especially for cases where we refer to the macroevolutionary history of Diplogastridae, where work has shown that the fixation of single morph has a correlation with morphological evolution, which we find an interesting use of the term "assimilation" and thus find useful to point out.

Lines 16-18: "Our findings thus point to the plausibility that modified spr-5 function sources morphological changes ahead of genetic change, giving mechanistic insight into how traits are modified as they traverse the continuum of greater to lesser environmental sensitivity."

Comment. An example of complex/convoluted wording that borders on the misleading (I refer to the "sources morphological changes ahead of genetic change" part: the work documents the effects of spr-5 mutants, and is not a test of a plasticity-first scenario or similar).

Our response: We have removed this overreaching statement.

Lines 24-27: "Beyond the fixation of allelic variants expected to occur during assimilation, and which have been identified in some cases^{5,6}, theory suggests that epigenetic mechanisms and nongenetic inheritance, because they might offer a bridge to genetically inherited change, should be key factors in facilitating genetic assimilation^{7–10}."

Comment: This is a bit awkward here in the Introduction, because the study was not originally set up to test for the role of epigenetic mechanisms and non-genetic inheritance. This line of investigation follows from the identification of spr-5 as a putative candidate (although there is arguably not much 'epigenetic' in this paper, as reviewer 2 notes). Arguably it is a matter of preference, but given the overall feeling that

there is a mismatch between how the paper is presented and what was done, this wording does come across as a bit too much ad hoc reasoning.

Our response: We appreciate this perspective and have changed the introduction to more general language that does not focus on epigenetics or nongenetic inheritance. Moreover, for clarity, we also include, in essence, the overview of the study's aims as summarized above by the reviewer.

Line 40. Spell out what the "signature of evolution associated with plasticity's loss" should be. Presumably relaxed selection?

Our response: We now write, "genes whose expression was once polyphenism-biased should bear a signature of evolution (e.g., relaxed or positive selection) associated with plasticity's loss."

Line 45-46. The use of genetic mutants to identify genes that are differentially expressed between morphs is creative, but it is also quite unexpected to not include wild-type morphs. The candidate genes identified here dictate all the downstream work, so it is peculiar to not demonstrate that these signatures are consistent with wild type phenotypes. Of course there is the risk of environmental effects 'blurring' the differences, but the two 'natural' morphs could still have been included since a given environment does not induce 100% of either morph. Note that, if one considers the lack of contrast here odd, the identification of genes under selection will come across as less-than-solid.

Our response: Addressing the "risk" acknowledged here, we already explain in the Results why we did not use individual environmental induction cues to produce morphs that we would sequence. Still, we agree that, *if technically feasible*, wild-type morphs would be ideal to include, or even be the only source of morph-biased transcriptomes. However, we have used genetic mutants (here and previously) because we have not conceived of another way to capture transcriptomes at the relevant developmental periods while also predicting what morph will ultimately be expressed, since there is some degree of stochasticity in the polyphenism decision (at least under laboratory conditions). In the wild-type, the only way to be sure that we are sequencing a given morph is after the nematodes' final molt, when the polyphenism is observable to the investigator, but also at the point when developmentally relevant processes are, presumably, mostly complete. Consequently, in this system, we do not agree that sequencing a pool of developing animals of unknown final phenotype (i.e., what adult morph they are developing to become), regardless of induction cues, or sequencing adults in which morph-specific gene networks have already been deployed would make our detection of morph-specific genes any stronger.

Lines 57-59. The authors will obviously be aware that transcriptomes of whole worms are associated with all kinds of problems when it comes to interpreting the results in terms of mouth morphology. I assume it was necessary to do this to get enough tissue. Yet, it would be good to keep in mind that any conclusions from this work needs to make the assumption that the relevant gene expression differences were identified. Some words of caution could be appropriate here or elsewhere (e.g., Discussion).

Our response: We have added the appropriate caveats to this section. As the reviewer guesses, tissue volume, which is to say, transcript abundance, is a limiting factor in our system. We hasten to add, however, that a resource polyphenism might be expected to also involve tissues beyond the mouth itself, such as the intestine, and a recent study from our lab has formally demonstrated this. The added text reads, "these genes were identified from pooled, whole-organism contrasts that, although they obscure developmental-stage or tissue-specific responses, we predicted to capture effects across the body, which are known for this case of resource polyphenism [Casasa et al. 2023 PNAS]."

Lines 66-70. Here it would be useful to say what morph *P. bucculentus* has, this helps to interpret the results later.

Our response: Added.

Line 72. Transcriptomes from the 41 species mentioned above, I assume?

Our response: Correct, and we have clarified this point.

Line 74: "...thereby defining evolutionary targets associated with the assimilation of once-plastic morphology"

Comment: Very awkward wording, why not just say loss of polyphenism?

Our response: We have made the change.

Lines 76-86: This whole section of results on the Diplogastridae are a bit difficult to make sense of and it would be helpful to include some details that help the reader see the difference between the two clades (transcriptomic vs genomic data; note that the latter is not actually 'genome-wide', or else it could not be compared to the former) and how the results are combined.

Our response: We have added more detail to this section and removed the term "genome-wide." The revised text reads:

"Specifically, we intersected our results to patterns of episodic, diversifying selection in the species *Diplogasteroides magnus* and *Levipalatum texanum*, whose mouth morphologies were, like the two *Pristionchus* species above, the product of increased evolutionary rates following assimilation. As in *P. elegans*, the assimilated forms of *D. magnus* and *L. texanum* have diverged beyond identification as either the Eu or St morph, although an outgroup to these species (*Oigolaimella* spp.) [Susoy et al. *eLife*] suggests the Eu morph may have been fixed. For these analyses, we used these two monomorphic species and polyphenic representatives from other genera with sequenced genomes (*Pristionchus*, *Parapristionchus*, *Micoletzkyia*, *Allodiplogaster*, and *Koerneria*) across the tree of Diplogastridae. Whereas our former analysis focused on closely related species within the genus for which the polyphenism-associated genes were inferred (*Pristionchus*), this analysis used distant species from across the family tree to explore the generalizability of the former's results. Twenty of those genes inferred to be polyphenism-biased in *Pristionchus* reported diversifying selection associated with both assimilation events (Table 1)."

Lines 86-89: "Consistent with the expectation that genetic assimilation, and the evolution of plasticity more generally, involves the accumulation of variants that modify, stabilize, and refine an adaptive plastic response^{2,24,25}, we found multiple loci to contribute to assimilation's evolutionary signature."

Comment: This is a very dense and convoluted way of saying that you identified several candidate genes. If I understand the authors logic, they did not strictly speaking identify any candidates at all, since the numbers are not greater than expected by chance (see comment and response to reviewer 2). The text does not make clear if the overlap between comparisons (i.e., clades and gene expression vs sequence data) are greater than expected by chance. Even if not, perhaps one can say that those that do overlap may show something interesting and therefore are candidates worth exploring (after all, we can never

know from the results of these analyses which candidates that are false positives, and the authors pursue a single gene based on its known function, not the strength of association). I believe some readers will take this issue quite seriously so it is not a good strategy to sweep it under the carpet.

Our response: We agree that our wording could be simplified and have now done so. Our experimental design, as the Reviewer suggests about candidates worth exploring, was to identify *significantly* polyphenism-biased genes with *significant* selection signatures (in both clades) that overlapped between the two types of selection analyses. It was not to test whether any category of genes was overrepresented in some contrast, although that would have been a result worth pursuing if positive (i.e., it would show that polyphenism genes are more likely to be selected upon in fixed-morph lineages, thereby testing general predictions about conditional expression in this system). Also, our previous wording was intended to contextualize our observations with the broader literature on plasticity. We now write, “although polyphenism-associated genes were not over-represented among genes showing evidence of selection (Extended Data Table 1), 20 genes reported diversifying selection associated with both assimilation events (Table 1). Therefore, this analysis identified a set of candidates whose function may contribute to plasticity regulation, morphological variation, or both.”

Lines 101-104: As pointed out in the SI (following a response to reviewer 2), “relaxed purifying selection and episodic diversifying selection are both consistent with positive shifts in ω .” This is true, but sadly it also means these numbers should be interpreted very carefully. Maybe in practice all that may have happened is just relaxed selection, for example? Overall, since it is unclear if the candidates derived from cross-validation (between clades and between gene expression and sequence data) are statistically supported candidates, one does wonder how much weight we should put on *spr-5* at this point...

Our response: Even if only relaxed selection has occurred, the gene’s function and theoretical arguments (e.g., Pál and Miklós 1999) suggest that it is a good candidate for further investigation. More generally, given that *spr-5* was in our polyphenism-associated gene set, has experienced significant sequence diversification associated with multiple losses of plasticity, has a demonstrated role in epigenetic regulation and nongenetic inheritance in other systems, and there is a body of literature suggesting that such genes and mechanisms contribute to the evolution and loss of plasticity, we believe that *spr-5* is an appropriate candidate for function analysis. We think that any one of these points could be justification for functionally evaluating this gene, but together the case is even stronger. To increase clarity of what we considered statistically supported, since all gene sets compared were significant in their polyphenism bias and selection signatures in both types of analyses, we give more detail about our design on pages 5 and 6 of the text.

Lines 115-118: Why use ancestral reconstruction – which must be tricky given fast episodic and relaxed selection – rather than compare to both species directly?

Our response: We used ancestral reconstruction so that we could compare the two species to the putative sequence from which they evolved and leverage the tool’s predictions of functional consequences for specific amino-acid substitutions. That is, the functional consequence predictions are *necessarily* based on changes from an ancestral sequence and thus could not be done for the two species directly.

Lines 123-125: Why is this interpreted as ‘complex evolutionary history’? Also, the evidence is consistent with relaxed selection, but is it also possible that you just detect the signatures of a large target of neutral sequence change?

Our response: We had defined “complex” by more than one type of selective pressure being possible over the lineages’ evolutionary history. However, we agree with the reviewer’s questioning of this imprecise phrasing, so we have removed this interpretation. And yes, it is possible that there is more neutral sequence change in the focal lineages, but that would be suggestive, given the phylogenetic component of the tests, of relaxation of selection intensity for specific amino acids. We now write, “together with our selection tests above, these results suggest that relaxed selective constraint potentially shaped much of this gene’s divergence or concomitant shift in function and that such relaxation could promote substantial changes in plasticity and morphology.” Also, it is worth noting here that relaxed selection on epigenetic modifiers in particular has been hypothesized to promote genetic assimilation, thus strengthening the case for focusing on *spr-5*.

Lines 136-138: “...with and without a *spr-5* loss-of-function allele, induction of the Eu morph, which was observed in *spr-5* wild-type individuals (estimate = 2.78, $Z = 7.19$, $P = 1.34 \times 10^{-12}$), was abolished in mutants (estimate = - 0.48, $Z = 1.91$, $P = 0.11$; Fig. 2B).”

Comment: This reporting stands out because the appropriate test is the morph ratio in the wild-type vs mutants, not separate tests for the two groups. The methods give the impression that the statistics were performed by comparing frequencies between wild-type and mutant, but it is not entirely clear how this was done, what was the level of replication, and so on. I encourage the authors to make sure it is transparent in the main text that the right groups have been compared (with other factors included in the analyses, if appropriate).

Our response: We appreciate this comment, which points out how we can improve the transparency of our statistical tests. As we had described in the Methods, but now also have added to the Results, we first assessed if there was a significant genotype-by-environment interaction, and only if so, did we *ad hoc* explore differences within genotypes. To summarize, yes, we compared wild-type to mutants and then reported here, that the interaction is driven by a difference in magnitude of change across treatment for the two genotypes. We now write, “as with culture conditions above, there was a significant genotype-by-environment interaction: when we starved a normally St-biased strain (RS5200B) with and without a *spr-5* loss-of-function allele, induction of the Eu morph, which was observed in *spr-5* wild-type individuals (estimate = 2.78, $Z = 7.19$, $P = 1.34 \times 10^{-12}$), was abolished in mutants (estimate = - 0.48, $Z = 1.91$, $P = 0.11$; Fig. 2B).”

Lines 145-148: “This uncovering of cryptic or new variation, especially in previously unoccupied regions of morphospace could, in principle, act as an evolutionary capacitor and fuel natural selection. In sum, mutations in *spr-5* affected the regulation and production of a morphology that has rapidly diversified in correlation with its genetic assimilation”

Comment. Again, the wording makes this sound much more complicated than it is. What this demonstrates is that knocking out the gene affects both the incidence of discrete morphs, and the morphology. Maybe this is unexpected (it does not seem like it would be?), but to say that ‘knocking out *spr-5* causes phenotypic variation’ is arguably easier to understand than the authors’ wording. Moreover, given what you are actually doing in the lab, it seems ambitious to extrapolate to consequences over evolutionary time...

Our response: We have simplified the wording as suggested. We agree that our extrapolation was perhaps too ambitious. Instead, to contextualize our findings in the wider body of plasticity literature, we write, “in short, knocking out this gene affects both the incidence of discrete morphs and the

morphology. More generally, mutations in *spr-5* affected the regulation and production of a morphology that has rapidly diversified in correlation with its genetic assimilation.”

Lines 153-155: “We reasoned that if *spr-5* could affect once-plastic morphology even after the completion of genetic assimilation, *spr-5* mutants should have an effect in *P. pacificus* individuals with morph-constitutive development.”

Comment: Why is this introducing a different species with a single morph? Why not use the ones that were the focal species in the first set of analyses?

Our response: If the reviewer is referring to the species that have assimilated a single morph in evolution (*P. bucculentus* and *P. elegans*), we think this is a sensible question, and our reason for using a different species was purely technical: gene knockouts are not yet possible in those other species. However, we did use the same species (*P. pacificus*) as we did for our tests of *spr-5* mutations’ effects on plasticity and morphology, which is directly relevant to our assay here. Additionally, we now point out that we use this particular line, as opposed to others available in *P. pacificus*, because it also has an integrated transgene and RFP reporter, which eased the identification of homozygous *spr-5* mutants that are also morph-constitutive. This detail has now been added to the methods: “After confirming homozygosity of the *spr-5* mutation and, by expression of the integrated RFP reporter, the presence of *eud-1* over-expression construct, we used geometric morphometrics...”

Lines 170-174: “We found that (i) heterozygous F1 showed a significant reduction in the Eu morph compared to a wild-type self-cross, (ii) this epigenetic effect was erased in the F2, which returned to the wild-type phenotype, and (iii) homozygous *spr-5* F3 clones were significantly less Eu- biased than homozygous wild-type F3, showing conversion to the mutant phenotype in exactly one generation.”

Comment: This is really quite complex (also, why is it an ‘epigenetic effect’ and not just a ‘maternal effect’?), and the summary is not accompanied by supporting statistics. Consider spelling things out in more detail to help the reader how and why the results lead to particular conclusions.

Our response: We agree that this section has a lot of detail to unpack. We chose the style of presenting the results verbally here, putting the statistics into the figure legend, with the intention of making it easier to follow and not be “choppy” with parenthetical breaks. We have now moved these details to the main text. Our choice of terminology for epigenetic effect was based on usage in the literature and the function of the gene, but we have now changed the wording to “intergenerational effect” to remain mechanistically agnostic on this point. Also, we do not use “maternal effect,” even if our experiments do not distinguish such an effect from other intergenerational effects, because we do not rule out whether fathers might also pass this information, which should be possible if written directly to chromatin.

Lines 165-170: What species is this now?

Our response: All of the functional assays occurred in a single species, *P. pacificus*. We have now clarified this at the beginning of the functional assay results, “we then directly tested the function of *spr-5* in the mouth polyphenism using the polyphenic model nematode *Pristionchus pacificus*.”

Lines 189-191: “Finally, we tested whether loss of *spr-5* function, and thereby the production of epimutations, would allow directional selection on the plasticity (i.e., morph induction) as well as new variants of the morphology itself.”

Comment: This statement contains many assumptions and connections that seem unnecessary and confusing. For one, as far as I can tell, you have not demonstrated that loss of *spr-5* function produces epimutations (maybe somewhere else or by others in *C. elegans*?). The second part of the sentence suffers from the general issues of convoluted wording that make everything sound much more complex than it is.

Our response: We now include references to studies in *C. elegans*, which have clearly shown the function of SPR-5 in producing epimutations (Katz et al. 2009 *Cell*; Greer et al. 2014 *Cell Reports*; Greer et al. 2016 *Cell Res*). The entire revised text is provided below.

Lines 191-194: "Since assimilated species, which show increased rates of form evolution, also show signatures of increased diversification in *spr-5*, we hypothesized that modified *spr-5* function could allow directional selection upon new morphologies."

Comment: This is really very difficult to follow.

Our response: We have rephrased the logic of our approach in this section:

"Finally, we tested whether loss of *spr-5* function, and presumably the production of epimutations [Katz et al. 2009 *Cell*; Greer et al. 2014 *Cell Reports*; Greer et al. 2016 *Cell Res*], would allow directional selection on plasticity (*i.e.*, morph ratio) and morphology. We were motivated by the possibility that short-term, nongenetic shifts in plasticity and morphology would offer one explanation for the correlation of *spr-5* selection with morphological divergence in Diplogastridae. Our reasoning was that weakened selection on epigenetic modifiers can promote phenotypic evolution [Pál and Miklós 1999 *J Theor Biol*] and even compromised function can be positively selected and adaptive, particularly in enzyme evolution [Bailey et al. 2003 *PNAS*]. Using *spr-5* loss-of-function mutants as a powerful proxy of this potential effect, we tested whether multigenerational shifts in mouth morphology were possible by selecting on the epigenetic variation mediated by *spr-5*..."

Lines 195-203: Sadly, this selection experiment seems to lack an appropriate control, something that reviewer 2 also picked up on. The authors dismiss the criticism by referring to previous work showing a failure to get a response to selection on these lines, but I am afraid that this misses the point. It is the risk of standing genetic variation (or new sequence variants, perhaps induced by the gene editing itself) in the present strain/population that is an issue. While we know that epigenetic variation exists, can be heritable, and enable responses to selection, it is still the case that demonstrating convincingly that epigenetic variation is the substrate for responses to selection will require appropriate controls (or sequencing of selected lines). As much as I would like to accept the assumption that all of this is epigenetic, in my opinion the authors would be better off were they to avoid such statements and simply refer to selection on the lines. That they do respond COULD mean that *spr-5* mutants accumulate heritable epigenetic variation, but demonstrating this will require further work (note that this does not undermine the value of the work as a whole, just again show a mismatch between results and inference in this paper).

Our response: We appreciate the reviewer's concern about formal controls, having constructively responded to a similar point in our last revision. We had originally assumed common appreciation for how negligible standing variation is in our laboratory model, hence our need to add more text justifying our study design, which we have now done. In *C. elegans*, a similar biological model with the same

breeding system as *P. pacificus*, the only way to select on genetic variation for a phenotype is to either introduce variation artificially (by mutagenesis) or collect variation into the same population through advanced intercrosses among separate (but still individually inbred) strains. We still maintain that (i) the previous study that used the same wild-type strain we use here (only now it has been subject to even more lab inbreeding) failed to drive phenotypic changes, (ii) the high levels of homozygosity that have been measured, and (iii) the several lines of evidence from work in *C. elegans* showing that *spr-5* mutants promote epimutations together point most probably to epimutations being the selective substrate. However, we agree that we cannot yet rule out the formal, if unlikely, possibility that *de novo* mutations, possible off-target effects from CRISPR/Cas9 editing, or genetic mutations caused by defective *spr-5* function contributed to the selectable variation in our *spr-5* mutant lines. Therefore, we accept the reviewer's suggestion to change our interpretation from the simple assumption that the changes were epimutations to more neutral wording. Also, we explicitly make our audience aware of our reliance on previous studies for designing the experiment and the rationale for our conclusions:

"...We tested whether multigenerational shifts in mouth morphology were possible by selecting on the epigenetic variation mediated by *spr-5*. For this test, we performed artificial selection on polyphenism phenotypes in highly inbred (*i.e.*, genetically homozygous) but likely epigenetically variable *spr-5*⁻ lines. Because selection on inbred lines of the wild-type background in which our mutants were made fail to drive any shift in morph bias [Bento et al. 2010 *Nature*], and given the self-fertilizing reproductive mode and high homozygosity of laboratory strains of *P. pacificus* [Morgan et al. 2012 *Mol Ecol*; Rödelsperger et al. 2014 *Genetics*], we would interpret any selectable differences over a short time to be most likely upon epimutations rather than standing genetic variation. Likewise, since the mutation rate of this species is typical of many animals (*i.e.*, on the order of 10⁻⁹) [Weller et al. 2014 *Genetics*], we consider it unlikely that any phenotype could detectably be selected from new mutations over a small number of generations."

Lines 202-207: Again, this reporting seems to imply that the authors did not conduct the appropriate statistical test of differences between groups. I hope they did.

Our response: Here we are reporting how each group changed over generations and the estimate values, along with the P-values, indicate the direction and significance of that change. Please see our response above for our rationale in choosing this design.

Discussion: Most of the additions are helpful, but please do consider the general comment about over-interpretation –extrapolation of results and context. It would be a shame if readers dismissed this work because of how the authors have chosen to present it.

Our response: We have dialed down our extrapolation in the Discussion, sticking to what we have directly shown, which we believe has improved the general presentation of our findings. Notably, we have removed our speculation on what *spr-5* might have done in lineages that have historically assimilated single morphs (*i.e.*, most of what was the third to last paragraph of the Discussion), since we do not test this directly. From that paragraph, we have only retained the speculation that plasticity- or morphology-relevant genetic mutations might result from *spr-5* defects, something that the Reviewer implies in challenging our assumption of epimutations above. We have moved this to an earlier paragraph, where we lay out explicitly what future work might do to test this.